



# Spatiotemporal variation of snow depth in the Northern Hemisphere from 1992 to 2016

Xiongxin Xiao[1, 2], Tingjun Zhang[1, 4], Xinyue Zhong[3], Xiaodong Li[1], Yuxing Li[1]

[1] Key Laboratory of Western China's Environmental Systems (Ministry of Education), College of Earth and Environment
Sciences, Lanzhou University, Lanzhou 730000, China

[2] School of Remote Sensing and Information Engineering, Wuhan University, Wuhan 430079, China

[3] Key Laboratory of Remote Sensing of Gansu Province, Cold and Arid Regions Environmental and Engineering Research
Institute, Chinese Academy of Sciences, Lanzhou 730000, China

[4] University Corporation for Polar Research, Beijing 100875, China.

*Correspondence to*: Tingjun Zhang (tjzhang@lzu.edu.cn)

**Abstract:** Snow cover is an effective indicator of climate change due to its impact on regional and global surface energy and
water balance, and thus also weather and climate, hydrological processes and water resources, and the ecosystem as a whole.
The overall objective of this study is to investigate changes and variations of snow depth and snow mass over the Northern
Hemisphere from 1992 to 2016. We developed a long term Northern Hemisphere daily snow depth and snow water equivalent
product (NHSnow) by applying the support vector regression snow depth retrieval algorithm, using passive microwave remote
sensing data from the period. NHSnow product was evaluated along with the other two snow cover products (GlobSnow and
ERA-Interim/Land) for its accuracy across the Northern Hemisphere. The evaluation results show that NHSnow performs
comparably well with relatively high accuracy (bias: -0.59 cm, mean absolute error: 15.12 cm, and root mean square error:
20.11 cm) when benchmarked against the station snow depth measurements. Further analyses were conducted across the
Northern Hemisphere using snow depth, snow mass, and snow cover days as indices. Analysis results show that annual average
snow mass has a significant declining trend, with a rate of about 19.72 km$^3$ yr.$^{-1}$ or a 13% reduction in snow mass. Although
spatial variation pattern of snow depth and snow cover days exhibited slight regional differences, they generally reveal the
decreasing trend over the most area of the Northern Hemisphere. Our work provides evidence that rapid changes in snow depth
and snow mass are occurring since the beginning of the 21$^{st}$ century, accompanied by dramatic climate warming.

## 1. Introduction

Seasonal snow cover is an important component of the climate system and global water cycle and has significant
impacts on the surface energy, hydrological processes and water resources, heat exchange between the ground surface and
the atmosphere, and the ecosystem as a whole (Immerzeel et al., 2010; Zhang, 2005; Robinson and Frei, 2000; Tedesco et al.,
2014). On account of the high albedo and low heat conductivity properties, snow cover may directly modulate the land
surface energy balance (Flanner et al., 2011), influence the soil thermal regime (Zhang et al., 1996; Zhang, 2005), and
indirectly affect the atmospheric circulation (Cohen et al., 2012; Li et al., 2018). Most regions in the Northern Hemisphere





rely on natural water storage provided by the snowpack (Diffenbaugh et al., 2013; Barnett et al., 2005), to supply water for domestic and industrial usage (Sturm, 2015; Qin et al., 2006). Therefore, accurate estimation and reliable information on snow cover spatial and temporal change at regional and global scales are critical to climate change monitoring, model evaluation, and water resources management (Brown and Frei, 2007; Flanner et al., 2011).

Snow depth (SD) is the most useful and commonly measured parameter at national meteorological and hydrological stations, and the sites for local and regional water resources assessment programs. Given the sparseness of measurements, it is impossible to fully capture the spatial variability of snow cover, especially at high altitude mountains and high latitude regions. Although in-situ observations can obtain accurate and relatively reliable SD/snow water equivalent (SWE) data, the time-consuming and cumbersome nature of the work means it is unrealistic in mountain regions and low population zones. Remote

sensing is an effective and powerful tool for obtaining quantitative information about snow cover over larger areas (Foster et al., 2011). Optical remote sensing has a high spatial resolution; however, clouds have become the greatest hurdle in obtaining optical satellite data for snow cover detection (Foster et al., 2011). Microwave remote sensing is the most attractive alternative to optical remote sensing at all times and under all weather conditions. Moreover, microwave remote sensing data can be used to estimate SD and SWE by providing dual-polarization information at different frequency channels (Chang et al., 1987; Che

et al., 2008; Takala et al., 2011).

    SD and SWE derived from passive microwave (PM) data have been widely applied to investigate regional and global climate change and validate hydrological and climate models (Brown et al., 2010; Brown and Robinson, 2011; Dai et al., 2017). Progress in satellite data acquisition and SD/SWE retrieval algorithm development has led to a global improvement in snow monitoring (Qin et al., 2006; Snauffer et al., 2016; Takala et al., 2011). The PM brightness temperature of the Scanning

Multichannel Microwave Radiometer (SMMR), Special Sensor Microwave Imager (SSM/I), Advanced Microwave Scanning Radiometer for Earth Observing System (AMSR-E), Advanced Microwave Scanning Radiometer 2 on the Global Change Observation Mission – Water (AMSR2), Special Sensor Microwave Imager (SSM/S), Special Sensor Microwave Imager Sounder (SSMIS), and the Fengyun-3 satellite B/C (FY-3B/C) are available, and several algorithms have been developed to estimate SD and SWE (Chang et al., 1987; Dai et al., 2012; Xiao et al., 2018; Pulliainen, 2006; Takala et al., 2011; Che et al.,

2008; Foster et al., 1997).

    Most retrieval algorithms operated on the principle that the difference in brightness temperature between 18 and 37 GHz reflects the quantity of SD/SWE (Chang et al., 1987). Over and underestimated trends are prevalent in the linear SD/SWE retrieval algorithms (Gan et al., 2013), for which there are two possible and reasonable explanations. One is that vegetation overlying snow attenuates the microwave signal emitted from ground and results in the underestimation of SD/SWE from PM

data (Che et al., 2016; Vander Jagt et al., 2013). To reduce the effect of tree canopy, a forest fraction was introduced into the retrieval algorithm developed to estimate SD/SWE (Foster et al., 1997; Che et al., 2008), or the retrieval algorithm was constructed based on particular land cover types (Goïta et al., 2003; Che et al., 2016; Derksen et al., 2005; Foster et al., 2009).



The other explanation is that the relationship between the snow properties (SD or SWE) and the PM brightness temperature is non-linear. New approaches (e.g., artificial neural networks, support vector regression, decision tree), which are intended to replace traditional linear methods, have emerged using data-mining and have been explored to retrieve SD and SWE (Gharaei-Manesh et al., 2016; Tedesco et al., 2004; Liang et al., 2015; Forman et al., 2013; Xue and Forman, 2015). However, there

remain some limitations for these retrieval algorithms due to the diversity of land cover types and the spatiotemporal heterogeneity of snow properties.

Numerous studies have reported the variation characteristics of snow cover at regional and hemispheric scales (Rupp et al., 2013; Dai et al., 2017; Derksen and Brown, 2012; Brown and Robinson, 2011; Huang et al., 2016). Huang et al. (2017) reported the impact of climate and elevation on snow cover variation in the Tibetan Plateau, including SWE, snow cover extent

(SCE), and snow cover days. Hori et al. (2017) developed a 38-year Northern Hemisphere daily SCE product and analyzed seasonal Northern Hemisphere SCE variation trends. SD, which provides an additional information to characterizing snow cover variation, is selected as the basis of analyzing spatiotemporal change of snow cover. Barrett et al. (2015) explored intra-seasonal variability in springtime Northern Hemisphere daily SD change in the phase of the Madden–Julian oscillation. Wegmann et al. (2017) compared four long-term reanalysis data sets with Russian SD observation data; however, this study

only focused on the snowfall season (October and November) and snowmelt season (April). SD change trends have also been analyzed at regional scales (Ye et al., 1998; Dyer and Mote, 2006). What's more, several studies also quantified the spatial and temporal changes in SWE or snow mass derived from satellite and reanalysis data. Specifically, Foster et al. (2009) reported the characteristics of seasonal SCE and snow mass in South America from 1979 to 2006. Mudryk et al. (2015) compared multiple data sets and examined the climatology of Northern Hemisphere SWE over the 1981 – 2010 period. Zeng et al. (2018)

analyzed the spatial and temporal variation characteristics of snow cover in the conterminous United States from 1982 to 2016 with annual maximum SWE and snow season as indices. Although the studies mentioned above have analyzed the variation characteristics of snow cover in the Norther Hemisphere, they mainly focused on the limited dimensionality of snow cover, i.e. only one or two snow cover variables were used.

There are, however, very limited data (station data, satellite data, or otherwise) that can provide both SD and SWE data

on a hemispheric scale. This paper describes an approach to develop a consistent 25-year dataset of daily SD and SWE of Northern Hemisphere utilizing multi-source data. The primary objective of this study is to develop hemispherical SD and SWE products (hereafter referred to as the NHSnow) over 25 years (1992 – 2016) with a 25-km spatial resolution using support vector regression (SVR) SD retrieval algorithm (Xiao et al., 2018). This paper will address the following questions: 1) How consistent are NHSnow and other sources snow cover data sets with in-situ SD observations? 2) What is the spatiotemporal

variability of SD and snow mass in the Northern Hemisphere from 1992 – 2016? Meanwhile, it is extremely challenging to make extensive quantitative validation of SD and SWE estimates.

This paper is organized into the following five sections. After the introduction section and literature review, section 2



describes the data sets used in this study. The methods of data pre-processing and snow cover products generation are explained in Section 3. Next, we describe NHSnow validation against in-situ snow observations, demonstrate the variability of snow cover in the Northern Hemisphere and discuss the potential controlling factors for the variations of snow cover utilized NHSnow data (Section 4). Finally, section 5 summarizes the work of this paper.

## 2 Datasets

### 2.1 Passive microwave data

Cloud often appears in snow cover regions during the winter and conceals snow cover from optical observations. This makes passive microwave remote sensing particularly advantageous for detecting snow cover. The SSM/I and SSMIS are PM radiometers on board the United States Defense Meteorological Satellites Program (DMSP) satellite (data available from the National Snow and Ice Data Center, http://nsidc.org/data/NSIDC-0032). The SSM/I (F11 and F13) datasets from this platform and SSMIS (F17) have been generated into the equal-area scale earth grid (EASE-Grid) with a 25-km resolution (Brodzik and Knowles, 2002; Armstrong, 2008; Wentz, 2013; Armstrong and Brodzik, 1995) (Table 1). The SCE and SD derived from F11 and F13 sensors data have high consistency, rendering the calibration between these two sensors for snow cover area and SD unnecessary (Dai et al., 2015). To minimize the melt-water effect, which can change the microwave emissivity of snow, only descending orbit (night-time) passive microwave data were used (Foster et al., 2009).

**Table 1.** Detail description for SSM/ and SSMIS sensors. H and V denotes horizontal and vertical polarization, respectively.

| Satellite | SSM/I | | SSMIS |
|---|---|---|---|
| Platform | F 11 | F 13 | F 17 |
| Temporal coverage | 1991.12-1995.5 | 1995.5-2008.6 | 2006.12 - |
| Channels (GHz) | 19 H, V; 22 V; 37 H, V; 85 H, V | | 19 H, V; 22 V; 37 H, V; 91 H, V |

### 2.2 Ground-based data

Daily ground-based SD measurements from two sources were used to construct and verify the SD retrieval model in this study. The first dataset is the Global Surface Summary of the Day (GSOD) dataset provided by the National Oceanic and Atmospheric Administration (NOAA) (https://data.noaa.gov/dataset/dataset/global-surface-summary-of-the-day-gsod). This online dataset, which was created in 1929, is derived from the Integrated Surface Hourly (ISH) dataset (Xu et al., 2016). There are fourteen daily elements in the GSOD dataset, including SD measured at 0.1 inch. The missing SD measurements or reported 0 on the day, were marked to 999.9. Data from approximately 30000 meteorological stations were recorded, of which more than 9000 were typically obtainable. In our study period and area, more than 17000 meteorological stations were selected with records from 1991 to 2016 (Fig. 1). All meteorological sites and stations are far away from large water bodies such as large



rivers, lakes, and oceans.

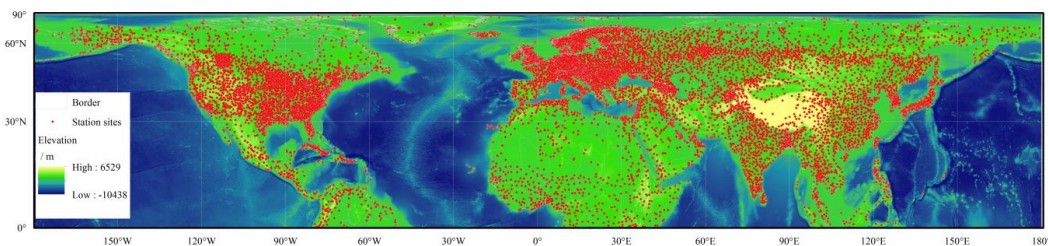

**Figure 1.** Distribution of Meteorological stations overlaid on ETOPO1 in the Northern Hemisphere.

To supplement station data that were not reported during the period 1992 to 2016, ground-based measurements of daily

SD were gathered from an additional 635 Chinese meteorological stations available at the National Meteorological Information

of China Meteorological Administration (http://data.cma.cn/en) (Xiao et al., 2018; Zhong, 2014). These daily records, collected

since 1957, include SD (unit, cm), observation time, and geographical location information.

**2.3 Topographic and land cover data**

We also used the topography as auxiliary information to estimate SD (Xiao et al., 2018). Elevation was available from

ETOPO1 at a resolution of 1 arc-minute (Amante, 2009) available at (http://www.ngdc.noaa.gov/mgg/global/). To match the

resolution of the PM brightness temperature data with 25-km spatial resolution, we resampled the ETOPO1 to 25-km resolution

(Fig. 1).

To increase the accuracy of SD estimates for different land cover types, we used both the MODIS land cover (MCD12Q1

V051) from 2001 to 2013 (Friedl and Sulla-Menashe, 2011; Friedl et al., 2010) and the Advanced Very High Resolution

Radiometer (AVHRR) Global Land Cover classification generated by the University of Maryland Department of Geography.

The MCD12Q1 International Geosphere-Biosphere Program (IGBP) classification scheme divides the land surface into 17

types, which were reclassified into five classes according to Xiao et al. (2018).

AVHRR imagery from the NOAA-15 satellite, acquired from 1981 to 1994 (Hansen et al., 2000), was categorized into

fourteen land cover classes at 1-km resolution. These data allowed us to adjust the proposed snow-depth retrieval algorithm

by reclassifying the fourteen native land cover classes into five classes (water, forest, shrub, prairie and, bare-land) at 25 km

spatial resolution (Table A.). MCD12Q1 is available at site https://earthdata.nasa.gov/, while AVHRR land cover data is

available online (http://www.landcover.org/data/landcover/).

**2.4 Snow cover datasets**

Two kinds of snow cover datasets were utilized, based on two criteria: covering the Northern Hemisphere, and long-term

availability. We selected GlobSnow and ERA-Interim/Land which are widely used in global and regional climate change
studies (Snauffer et al., 2016; Hancock et al., 2013; Mudryk et al., 2015). These datasets were compared with the NHSnow SD product.

In November 2013, the European Space Agency (ESA) released the GlobSnow Version 2.0 SWE and Snow Extent (SE) data for the Northern Hemisphere (Takala et al., 2011; Pulliainen, 2006). These data include all non-mountainous areas in the

Northern Hemisphere and are available online (http://www.globsnow.info/). Processing includes data assimilation based on combining satellite PM remote sensing data (SMMR, SSM/I and SSMIS), spanning December 1979 to May 2016, with ground-based observation data in a data assimilation scheme to derive SWE. GlobSnow Version 2.0 (hereinafter referred as GlobSnow) provides three kinds of temporal aggregation level products with 25 km spatial resolution: daily, weekly, and monthly. This dataset covers all land surface areas in a band between 35° N ~ 85° N, excluding mountainous regions, glaciers, and Greenland.

To convert between SD and SWE using GlobSnow, the snow density is held constant at 0.24 g/cm$^3$ (Sturm et al., 2010; Hancock et al., 2013; Che et al., 2016).

ERA-Interim/Land (Balsamo et al., 2015) is a global land-surface reanalysis product with data from January 1979 to December 2010 based on ERA-Interim meteorological forcing. It is produced by a land-surface model simulation using the Hydrology Tiled ECMWF Scheme of Surface Exchange over Land (HTESSEL), with meteorological forcing from ERA-

Interim. Dutra et al. (2010) described the snow scheme and demonstrated the verification using field experiments. SWE, which is labelled as SD in this dataset, is one of the thirteen parameters provided. We converted SWE to SD using the associated snow density data. These two datasets are available online (http://apps.ecmwf.int/datasets/data/interim-land/type=an/). The time of this reanalysis data with the analysis type used in this study is to maximize the proximity to the descending orbit time of the passive microwave sensor, and the spatial resolution of these reanalysis data is 0.125 degree.

**2.5 Snow classification data**

To accurately estimate SWE, snow classification data were used to convert SD into SWE. Global Seasonal Snow Classification System was defined by Sturm et al. (1995) based on snow physical properties (SD, thermal conductivity, snow density, snow layers, degree of wetting, etc.) and seasonal snow cover. Snow cover was categorized into six snow classes (tundra, taiga, alpine, maritime, prairie, and ephemeral) plus water and ice fields (Fig. 2). Snow classification data can be

accessed from the National Center for Atmospheric Research /Earth Observing Laboratory (https://data.eol.ucar.edu/dataset/6808). The snow classification dataset was developed and tested for the Northern Hemisphere at 0.5-degree spatial resolution (Sturm et al., 1995).

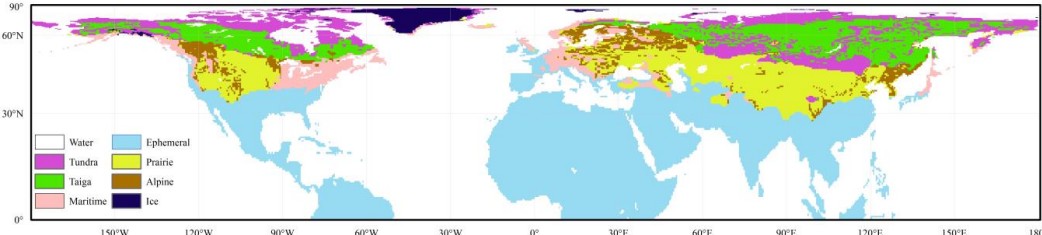

**Figure 2.** Snow Class distribution in the Northern Hemisphere

## 3 Methods

### 3.1 Theoretical basis

Snow distribution is affected by various factors, which include, but are not limited to: vegetation (Che et al., 2016; Vander Jagt et al., 2013), soil and air temperature (Forman and Reichle, 2015; Grippa et al., 2004; Dai et al., 2017), and topography and wind (Smith and Bookhagen, 2016; Dong et al., 2014). The snow retrieval process uses a range of variables to yield snow parameters (Eq. 1) (Xiao et al., 2018).

$$S = g\,(A, T, G, L, DS, D\, ...) + \varepsilon \tag{1}$$

where S is the snowpack properties (e.g., snow grain size, SD). g ($\cdot$) denotes the retrieval function. A is the atmosphere factors

(e.g., wind speed, air temperature). T is the topography factors (e.g., elevation, slope, aspect). G is the ground surface environment factors (e.g., surface temperature, vegetation type). L is the location factors (latitude, longitude). DS is the digital signal from the remote sensing sensor (e.g., PM, optical remote sensing). D is the time factor and $\varepsilon$ is the residual error or uncertainty (the difference between the sensor observation and the ground measurement). This retrieval formula can be exemplified by a general research in which SD variable (S) us usually derived from PM data (DS) using a linear regression

method (g($\cdot$)) (Chang et al., 1987)

### 3.2 Processing flow overview

Xiao et al. (2018) developed the SVR SD retrieval algorithm, which used a non-linear regression method (SVR) as the retrieval function (g ($\cdot$) in Eq. 1). Following Eq. 1, we used ten variables as the inputs, including PM brightness temperature (19 GHz, 37 GHz, 85 or 91 GHz) with vertical and horizontal polarizations, geographical location (latitude and longitude),

elevation, and the measured SD. The SVR SD retrieval algorithm also indirectly considered seasonal variation (day of the year; D) and vegetation (land cover; L) influences in the evolution of snow properties to improve the accuracy of the estimated SD. The output parameter is the estimated SD. In the Eurasia region, the SVR SD retrieval algorithm performed well with reduced uncertainties compared to measured data from ground stations, based on the correlation coefficient (R), mean absolute error (MAE), and root mean squared error (RMSE) (Xiao et al., 2018). It should be noted that this study used daily observation in



the Northern Hemisphere, except for July and August. The SVR SD retrieval algorithm mainly involves six steps (Fig. 3). In this study, we briefly describe the unchanged steps (steps 1, 2, 4 and 5) for the SVR SD retrieval algorithm and provide more detailed information for the changed (steps 3 and 6) and additional steps (steps 7 and 8).

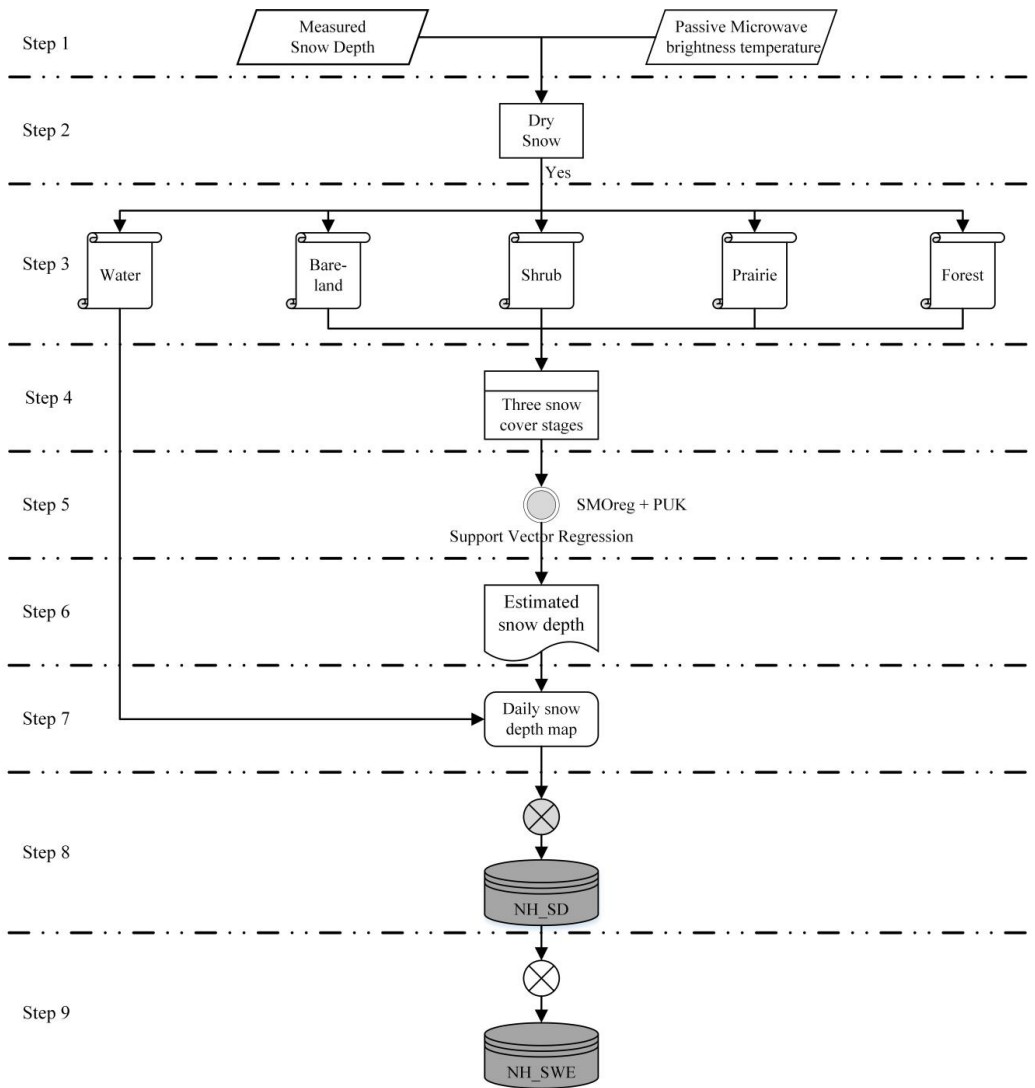

**Figure 3.** Process flowchart diagram for developing Northern Hemisphere daily snow depth and snow water equivalent data

Step 1 involves pre-processing meteorological station SD measurements and PM brightness temperature data. Before estimating SD from PM data, it is necessary to identify snow cover and dry snow by a set of criteria involved in Grody's algorithm (Grody and Basist, 1996) (in step 2).

In step 3, we tried to segregate the land cover effect on developing the SD retrieval models. For the years over which our study period pre-dates MODIS data, we used AVHRR land cover as supplement data. We reclassified MODIS and AVHRR





land cover into four classes (forest, shrub, prairie, and bare-land) which were bases for constructing the SD retrieval sub-model.

Table A (in the Appendix) describes the reclassification scheme for AVHRR land cover. MODIS land cover reclassification

schemes were documented by Xiao et al. (2018). Because of the relatively stable change of land cover, MODIS land cover in

2013 was used for each year from 2013 to 2016. Similarly, MODIS land cover in 2001 was used for each year from 1998 to

2001, and AVHRR land cover data was used for each year from 1992 through 1997. The retrieval sub-models were established

on different land cover types (forest, shrub, prairie, bare-land).

We converted the day of the year (D) into three snow cover stages: snow accumulation stage, stabilization stage, and

ablation stage. To consider the evolution effect of snow properties (step 4), the SD retrieval model was established on the three

snow cover stages, respectively. In step 5, we chose SVR as the retrieval function (Eq. 1) with specific kernel functions and

parameters.

Step 6.1. Construction of a sub-continental model. In this study, we separately constructed the SD retrieval models for

Eurasia and North America, based on the information that the snow properties show a discrepancy between Eurasia and North

America. Taking snow density as an example, Bilello (1984) pointed out that mean snow density in the former Soviet Union

($0.21 \sim 0.31$ g/cm$^3$) was lower than that in North America ($0.24 \sim 0.31$ g/cm$^3$). Zhong et al. (2014) explained the reasons for

such a difference between Eurasia and North America.

**Table 2.** Training sample filter rules

| Group ID | Filter rules |
|---|---|
| group2. | If $\text{Number}_{total}(group2) \leq 3000$<br><br>$\qquad\qquad\qquad \text{Number}_{training}(group2) = \left(\text{Number}_{total}(group2)\right)/2$<br><br>Else $\text{Number}_{training}(group2) = 3000$ |
| group3. | If $\text{Number}_{total}(group3) \leq 3000$<br><br>$\qquad\qquad\qquad \text{Number}_{training}(group3) = \left(\text{Number}_{total}(group3)\right)/2$<br><br>Else $\text{Number}_{training}(group3) = 3000$ |
| group1. | If $\text{Number}_{training}(group2) > 2000$ or $\text{Number}_{training}(group3) > 1000$<br><br>$\qquad \text{Number}_{training}(group1) = 15000 - \text{Number}_{training}(group2) - \text{Number}_{training}(group3)$<br><br>Else $\text{Number}_{training}(group1) = 12000$ |

Step 6.2. The selection of training sample. The accuracy of the estimated SD primarily depends on the quality of the

training samples (Xiao et al., 2018). More data than needed in the training stage to train the SD retrieval model may lead to

over-fitting and yield the estimated SD with a high error. In this study, we collected numerous daily records for over 25 years.

A changed sample selection rule was used in this study to avoid data information redundancy and was divided into two steps.

First, the number of samples of the three groups split by snow depth values should be solidly quantified, i.e. group1 (0 cm $\leq$



SD < 50 cm; shallow snow), group2 (50 cm ≤ SD < 100 cm; intermediate depth) and group3 (SD ≥ 100 cm; deep snow). To avoid an inflated training sample in group2 and group3, we set a threshold (3 000) determined by several tests (not shown). A threshold (12000) for group1 was adopted following Xiao et al. (2018). Table 2 details the selection rules for the training sample for each group. Second, the quality of the training sample in each group was controlled using stratified random sampling.

Stratification was performed at 1 cm SD intervals. All selected operations at this step were based on random selection of samples.

Step 7. Through the above steps, it generally created 24 SD retrieval sub-models (2 (continents) * 4 (land classes) * 3 (stages)) for producing the daily estimated SD data in the Northern Hemisphere from January 1992 to December 2016 (excluding July and August). Owing to radiometer observations, NHSnow products are only reliable in areas with seasonal dry

snow cover. Areas with sporadic wet or thin snow are not reliably detected, and areas marked as snow-free may include areas with wet snow. If one pixel is detected as snow cover by the detection decision tree (Grody and Basist, 1996), but is likely to be shallow snow with an estimated value of equal or less than 1 cm, the SD value is set as 5 cm (Che et al., 2016; Wang et al., 2008) (Fig. 4).

Step 8. In this study, Greenland and Iceland are excluded from the generation and analysis of NHSnow (NH_SD,

NH_SWE) products because of the difficulty in discriminating snow from ice (Fig. 4) (Brown et al., 2010). Missing data and zero-data gaps occur in generating daily SD grid products. Therefore, we applied the following filter: the daily estimated SD was defined as the midpoint of a sliding 7-day average window to reduce noise and compensate for missing data in the daily time series. For example, the SD estimate for 4 January is an average of the assimilated scheme output for 1 to 7 January (Takala et al., 2011; Che et al., 2016). When finished, the sliding SD method generated daily SD products for the entire

Northern Hemisphere (NH_SD; Fig. 4).

**3.3 Estimation of SWE**

SWE contains more useful information for hydrologists than SD because it represents the amount of liquid water in the snowpack when the snow melts. It is a great challenge to determine the precise distributions of SWE at regional and global scales (Chang et al., 1987; Kongoli, 2004; Tedesco and Narvekar, 2010; Bair et al., 2018). Snow density, which can be used to

convert SWE from SD, is a key factor in accurately estimating SWE (Sturm et al., 2010; Tedesco and Narvekar, 2010). Northern Hemisphere SWE products were generated using snow density ($\rho_{snow}$) that converts SD to SWE (Eq. 2; Step 9 in Fig. 3; Fig. 4).

$$\text{SWE}(mm) = \text{SD}(cm) \times \rho_{snow}(g/cm^3)/\rho_{water}(g/cm^3) \times 10 \qquad (2)$$





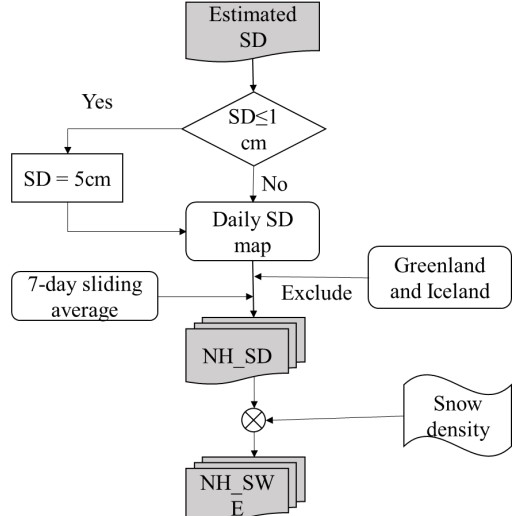

**Figure 4**. Flowchart diagram of the generation of NHSnow products.

At present, the primary problem is to obtain a relatively accurate snow density. Two methods are usually used to convert SD to SWE. The first method uses a fixed value of snow density (0.24 g/cm$^3$, or another value), without spatiotemporal variation (Che et al., 2016; Takala et al., 2011). The second uses a temporally static but spatially variable mask of snow density to estimate SWE (e.g., AMSR-E SWE products) (Tedesco and Narvekar, 2010). Since snowpack is usually rather unstable, it is unrealistic to set the snow density for the whole snow season to a constant. Observations show that snow density evolves and increases throughout the snow season (from September to June) (Dai et al., 2012; Sturm et al., 1995). Snow density typically varies from 0.05 g/cm$^3$ for fresh snow at low air temperatures to over 0.55 g/cm$^3$ for a ripened snowpack (Anderton et al., 2004; Cordisco et al., 2006). Therefore, a dynamical calculation method was adopted to estimate daily snow density, which is obtained following Eq. 3 with the assumption that snowpack occurs as a single layer (Sturm et al., 2010). Daily SD, the day of the year (DOY), and the snow climate class (SCC) were used to produce snowpack bulk density estimates. In this method, knowledge of SCC is used to capture field environment variables (air temperature, initial density) that have a considerable effect on snow density evolution.

$$\rho(SD, DOY, SCC) = (\rho_{max} - \rho_0)[1 - exp(-k_1 \times SD - k_2 \times DOY)] + \rho_0 \qquad (3)$$

where $\rho_{max}$ is the maximum density, $\rho_0$ is the initial density, $k_1$ and $k_2$ are densification parameters for SD and DOY, respectively. $k_1$, $k_2$, $\rho_{max}$, $\rho_0$ vary with SC (Table 3). For operational purposes in this work, DOY extends to 1 September each year (Matthew Sturm, personal communication, 2018) running from -122 (1 September) to 181 (30 June). Sturm et al. (2010) did not provide snow density for the SCC of ephemeral snow despite its presence in the Northern Hemisphere. According to Zhong et al. (2014), 0.25 g/cm$^3$ was taken as the mean density of ephemeral snow with no evolution throughout the snow cover year. Finally, daily snow density is simulated by the Eq. 3 during the 1992−2016 period.



**Table 3.** Snow density estimation model parameters

| Snow class | $\rho_{max}$ | $\rho_0$ | $k_1$ | $k_2$ | References |
|---|---|---|---|---|---|
| Alpine | 0.5975 | 0.2237 | 0.0012 | 0.0038 | |
| Maritime | 0.5979 | 0.2578 | 0.0010 | 0.0038 | |
| Prairie | 0.5940 | 0.2332 | 0.0016 | 0.0031 | Sturm et al. (2010) |
| Tundra | 0.3630 | 0.2425 | 0.0029 | 0.0049 | |
| Taiga | 0.2170 | 0.2170 | 0 | 0 | |
| Ephemeral | 0.2500 | 0.2500 | 0 | 0 | Zhong et al. (2014) |

## 4 Results and Discussion

### 4.1 Snow depth

#### 4.1.1 Validation of snow depth

5   To provide insight into the relative performance of SD products, we compared three sources of snow cover data sets (NHSnow, GlobSnow, and ERA-Interim/Land) with ground SD measurements (Fig. 5-7) using three indices (bias, MAE, and RMSE). We collected daily SD measurements for the common period (1992 – 2010) of the three products as validation data. This primarily focuses on the snow stabilization stage (December to February). Since the snow density changes slowly over a smaller range in the snow cover stabilization stage (Xiao et al., 2018), using a constant value (0.24 g/cm³) for GlobSnow

10  (Section 3.3). Subject to the unavailability of SWE station observations, the evaluation of SWE could not be carried out.

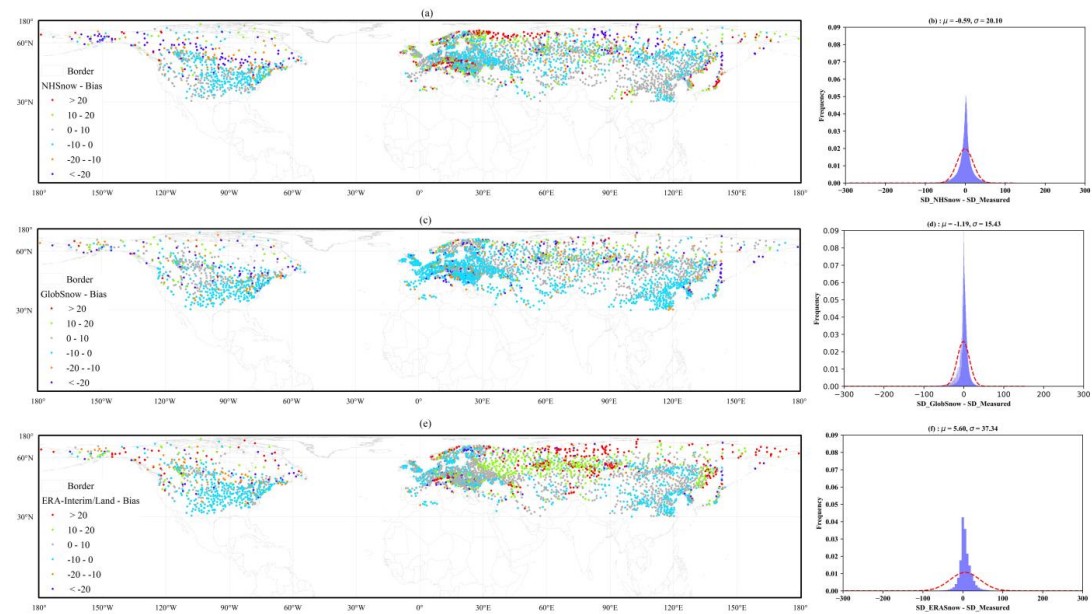





**Figure 5.** Bias of each meteorological station and histogram of biases for three products: a), b) NHSnow; c), d) GlobSnow, e), f) ERA-Interim/Land. The red dashed line in right column figures are the fitted normal distribution curve

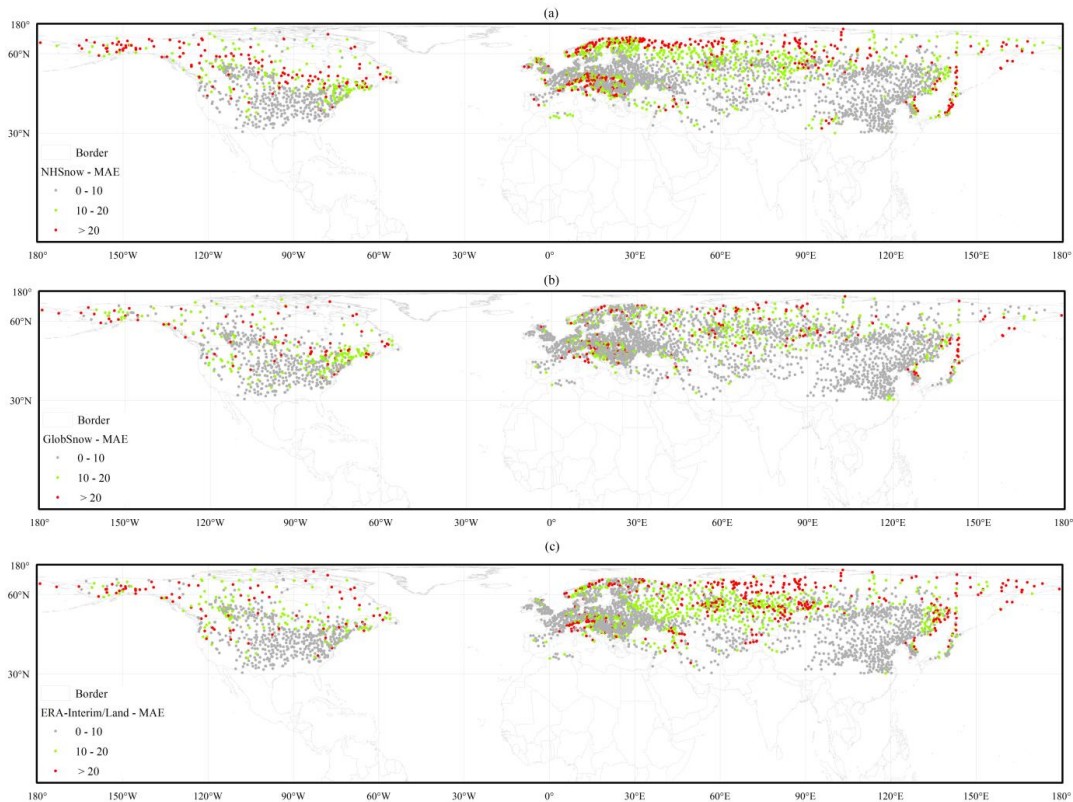

5     **Figure 6.** MAE of each meteorological station for three products: a) NHSnow, b) GlobSnow, c) ERA-Interim/Land.

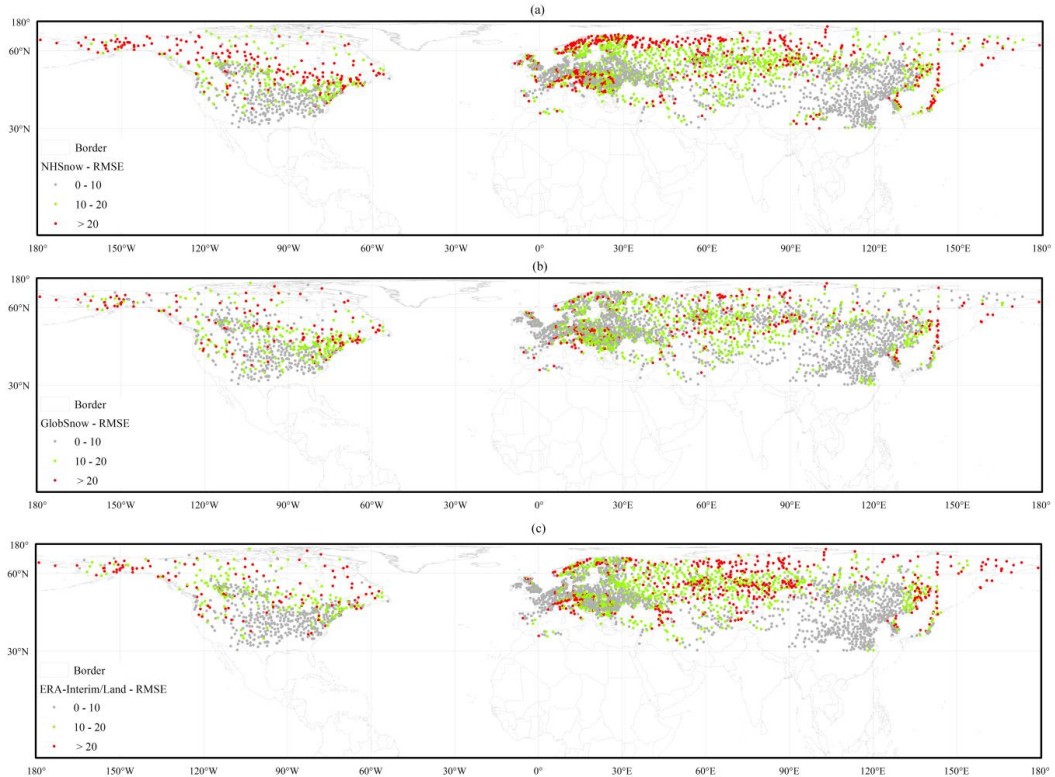

**Figure 7.** RMSE of each meteorological station for three products: a) NHSnow, b) GlobSnow, c) ERA-Interim/Land.

A small bias (blue and green dots, Fig. 5) between the estimated SD against measured SD is found for the mid and low

latitude regions (< 60 °N) for the three SD datasets (NHSnow, GlobSnow, and ERA-Interim/Land). However, a large bias was

found for the polar region and along the coast, such as the Russian coastal regions, the Russian Far East, the Korean peninsula

region, and Northeast Canada. For NHSnow and GlobSnow, most bias is distributed near the μ=0 line with high frequency,

although some bias is greater than 100 cm (or less than −100 cm) (Fig. 5b, d). Positive (negative) biases indicate that the mean

estimated values are greater (less) than the corresponding measured SD values. ERA-Interim/Land overestimated snow depth

in Western Siberian Plains and Eastern European Plains (around 60 °N; red nots, Fig. 5e). As a reference, the average SD

pattern of three products in February (1992 – 2010) were also provided in the Appendix (Fig. A)

For analysis indexes, MAE and RMSE, the distribution of error points of NHSnow and GlobSnow are much the same as

the distribution of its bias (Fig. 5 – 7). We used all evaluation records to calculate three precision indexes for three products,

and found that the bias, MAE, and RMSE are -0.59 cm, 15.12 cm, and 20.11 cm, respectively, for NHSnow grid products. But

for GlobSnow, there is more bias (-1.19 cm), MAE (15.98 cm) and lower RMSE (15.48 cm) (Table 4). This comparison

(NHSnow vs. GlobSnow) showed relatively good agreement, although NHSnow over- or underestimated the SD with larger

RMSE. Overall, the performance of GlobSnow was better than the NHSnow grid product. However, part of the validation data

was also applied for GlobSnow assimilation; it is highly possible that in this case, GlobSnow validation may not completely



independent. The different performance for these two products may be mainly caused by the evolution of snow grain size was used by HUT (The Helsinki University of Technology) model to generate SWE in GlobSnow. Che et al. (2016) reported that the grain size is more important than snow density and temperature. Further, ERA-Interim/Land had the worst performance of all three products with the highest bias (5.60), MAE (18.72) and RMSE (37.77). The smallest bias is found for the mid-latitude regions (< 50 °N) and much of the bias lay at 0–100 cm for ERA-Interim/Land products (Fig. 5e, f). We can find large MAE and RMSE in high latitude and coastal regions (Fig. 5e). Unlike NHSnow and GlobSnow, ERA-Interim/Land is more likely to overestimate SD and appears to be less consistent with in situ observations across the Northern Hemisphere (Fig. 5f). Through analyzing ground observation, we can see that deep snow is distributed in high latitude areas.

While these grid products did a fairly good job in small SD accumulations (shallow and mid-deep snow cover) regions, they all struggle to capture the SD with low bias, MAE, and RMSE in very high SD accumulations (deep snow) regions (Fig. 5 – 7, Fig. A). As a result, snow cover variation could not be adequately captured in areas with deep snow; thus, we should be cautious when interpreting the validation result in deep snow regions.

**Table 4.** The evaluated indexes (bias, MAE, RMSE; unit: cm) for three grid SD products (NHSnow, GlobSnow, ERA-Interim/Land).

| Products | Bias | MAE | RMSE |
|---|---|---|---|
| NHSnow | -0.59 | 15.12 | 20.11 |
| GlobSnow | -1.19 | 15.98 | 15.48 |
| ERA-Interim/Land | 5.60 | 18.72 | 37.77 |

Ground temperature and topographic factors could caused a great discrepancy between measured and estimated SD (Vander Jagt et al., 2013; Snauffer et al., 2016). Forests also exhibit a strong influence on snow redistribution and the evolution of snow properties. Dense portions of boreal forests are widely distributed in North America and northern Eurasia (Friedl et al., 2010). In the vast areas covered with tall vegetation (forests and shrub), the errors in SD estimates are quite large (Fig. 5-7). Furthermore, spatial inhomogeneity makes it impossible for a grid cell (~25 km) to be completely covered by one vegetation type (low heterogeneity). Because the estimated SD of NHSnow depends on land cover types, this discrepancy induced by surface cover heterogeneity could partly account for why NHSnow has a smaller MAE and RMSE for low vegetation (bare-land and prairie) distributed at middle and low latitudes (Xiao et al., 2018).

There are scale mismatches between in situ observation and the grid products regarding snow properties and their spatiotemporal representativeness (Frei et al., 2012). It is difficult to accurately validate the observation results of coarse-resolution satellites using ground measurements. Subsequently, over- or underestimates are inevitable when using a single in situ (SD or SWE) observation to test the veracity of the grid products (Mudryk et al., 2015; Xiao et al., 2018). Snow surveys would benefit from multiple measurements at different points within one pixel (López-Moreno et al., 2011). In situ observations are highly representative when the SD varies smoothly in space and poorly representative when the SD exhibits





large fluctuations (Che et al., 2016). However, there is almost always a lack of sufficient ground-measured data. To date, field site observations are still more authentic and reliable than satellite datasets. As a whole, the accuracy of the estimated SD in the Northern Hemisphere presented a spatial heterogeneity. Issues of scale and spatial heterogeneity of validation data notwithstanding, these comparisons conducted in our work can yield valuable insight into the performance of these products.

5    **4.1.2 Variation of snow depth**

To better understand and interpret snow cover variation in the Northern Hemisphere, we analyzed the SD variation using seasonal maximum SD from 1992 to 2016. According to the rules of variation trend grading, the SD variation was divided into 5 grades (extremely significant decrease, significant decrease, non-significant change, significant increase, and extremely significant increase; Table 5). A least-squares regression was used to analyze the variation of snow cover property for each

10    pixel, with a per-pixel evaluation of significance level (F-test).

**Table 5.** Rules of variation trend grading

| Variation rate | P value | Variation level |
|---|---|---|
| rate ≪ 0 | $p \leq 0.01$ | extremely significant decrease |
| rate < 0 | $0.01 < p \leq 0.05$ | significant decrease |
| - | $P > 0.05$ | non-significant change |
| rate > 0 | $0.01 < p \leq 0.05$ | significant increase |
| rate ≫ 0 | $p \leq 0.01$ | extremely significant increase |



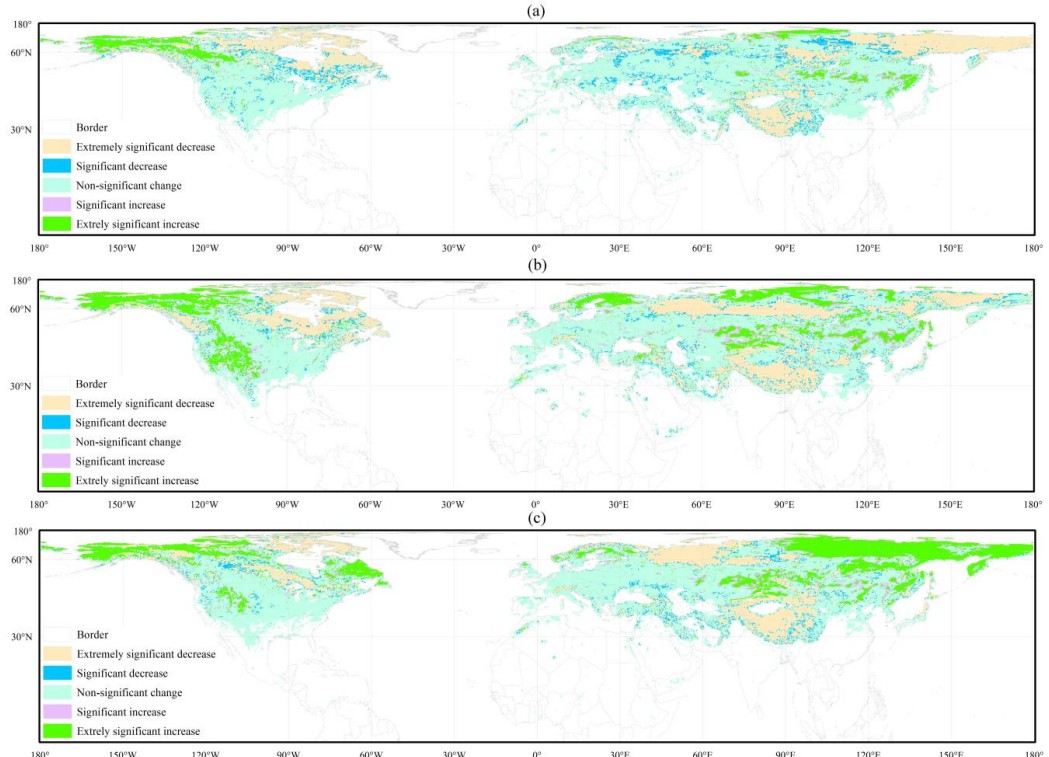

**Figure 8.** The variation rate pattern of season maximum SD with statistical significances over the Northern Hemisphere for three snow cover season, fall (a; September to November), winter (b; December to February), spring (c; March to June) from 1992-2016.

Fig. 8 illustrates the variation pattern of seasonal maximum SD in three seasons (fall, winter, and spring) with a statistical

significance level. In three seasons, the variation trend of seasonal maximum SD exhibits a distinctly different pattern in the

Northern Hemisphere since 1992. Fig. 8a presents the patterns of the change trend of maximum SD. We found that the areas

which exhibit the extremely significant decrease trend in fall (Fig. 8a) are mainly located in the Russian Far East, the Qinghai–

Tibet Plateau, the southern Siberian Plateau, and the northeastern regions of Canada. On the other hand, Russia's Taimyr

Peninsula and the Alaska region exhibit the extremely significant increase trend (0 – 1 cm yr.$^{-1}$). Moreover, the maximum SD

in winter and spring also undergoes the extremely significant decrease trend in the Qinghai–Tibet Plateau and the northeastern

region of Canada (Fig. 8b and 8c). The areas with extremely significant decrease extend to the Western Siberian plain region.

Wang and Li (2012) used nearly 50 years of daily station SD observation data to analyze the trend of maximum SD in China.

The variation trend of seasonal maximum SD in the Qinghai-Tibet Plateau reported by Wang and Li (2012) is consistent with

the observation in this study. There are more regions which show the extremely significant increase trend in winter and spring

(green regions). The seasonal maximum SD variations in fall and winter in Russian Far East exhibit the extremely significant

decrease trend, while in spring, it shows an extremely significant increase trend. This variation trend of maximum SD in spring,

analyzed using NHSnow products, is consistent with the analysis results obtained using GlobSnow products (Wu et al., 2018),

showing increasing trends in the Scandinavian peninsula and Alaska regions, and decreasing trends in the Kamchatka Peninsula



region.

Finally, we analyzed seasonal variation of SD across the Northern Hemisphere using seasonal average SD. Seasonal average SD is defined as the cumulative SD divided by the days in one snow cover season (refer to Eq. A in Appendix). SD variation rate fluctuates in different regions and seasons. The rate is generally large in the region north of 55° N (Fig. 9, Fig.

B and C in appendix). This fluctuation is large in winter with a higher rate of −0.11 ± 0.40 cm yr.$^{-1}$ than other seasons during the 1992–2016 (Fig. 9d, Table 6.), which implies that the maximum changes in average SD occurred in winter. A similar conclusion can be found in the two periods 1992–2001 and 2002–2016 (Fig. B-d, C-d in the Appendix; Table 6). Although not all variation trends passed the significance test, most regions in the Northern Hemisphere show increasing trends during the 1992-2001 (Fig. B; Table 6). The SD variation trend in the three seasons during the 2002–2016 is reversed. The absolute

variation rate during the 2002–2016 is apparently greater than that during the 1992–2001 (Fig. C; Table 6). We found that the high fluctuation of the SD variation rate especially occurred in the Arctic and the Qinghai-Tibetan plateau for three seasons. The selection of this time breakpoint (2001/2002) was because it is the beginning of the new century.

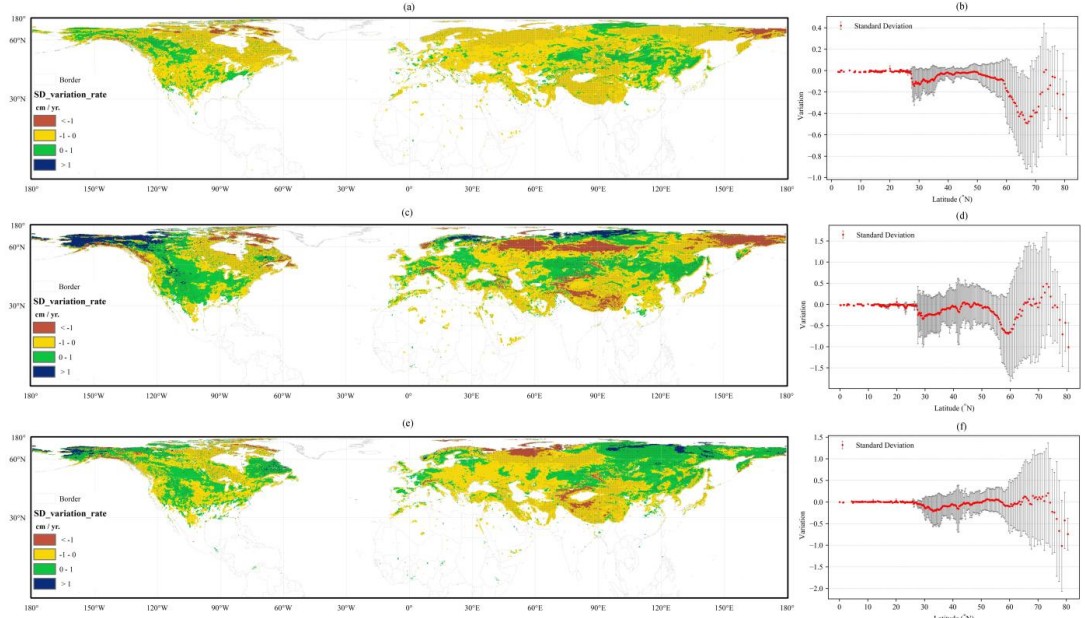

**Figure 9.** The variation rate pattern of season average SD over the Northern Hemisphere for three snow cover season, fall (a,

b; September to November), winter (c, d; December to February), spring (e, f; March to June) from 1992-2016. Black dots in (a, c, e) indicate that the changes are significant at 95% confidence level (CL). The zonal distributions (b, d, f) are mapped at a 0.25-degree resolution in latitude. The error bars in (b, d, f) is one time of standard deviation.

**Table 6.** Mean variation rate of average SD (cm yr.$^{-1}$) over the Northern Hemisphere for three common period (1992-2016, 1992-2001, 2002-1996) and snow cover seasons (fall, winter, spring). Std. means standard deviation





| Season | 1992-2016 (Mean ± 1 Std.) | 1992-2001 (Mean ± 1 Std.) | 2002-2016 (Mean ± 1 Std.) |
|---|---|---|---|
| Fall | -0.08 ± 0.11 | -0.01 ± 0.19 | -0.15 ± 0.22 |
| Winter | -0.11 ± 0.40 | 0.06 ± 0.62 | -0.22 ± 0.75 |
| Spring | -0.04 ± 0.25 | 0.02 ± 0.51 | -0.07 ± 0.41 |
| Year | -0.06 ± 0.20 | 0.02 ± 0.35 | -0.11 ± 0.34 |

**4.2 Snow mass**

The GlobSnow dataset covers all land surface areas excluding mountainous regions, glaciers, and Greenland. From the above analysis, we found that ERA-Interim/Land has a somewhat poor performance in term of SD estimation. Therefore, we took NHSnow products as the analysis data to further analyze the variations of snow mass in the Northern Hemisphere. Snow

5 mass is calculated by SWE multiplied by the snow cover area (Qin et al., 2006). We should note that the snow classification tree (Grody and Basist, 1996), which has been applied in many studies (Che et al., 2008; Dai et al., 2017; Yu et al., 2012), was used to detect snow cover for the NHSnow product. Liu et al. (2018) also reported that Grody's algorithm has higher positive predictive values and lower omission errors by evaluating the snow cover mapping algorithms using the in-situ SD over China. In this section, the annual maximum, the annual average snow mass and the monthly average snow mass in each snow cover

10 year (from September 1 through June 30) were calculated over 25 years.

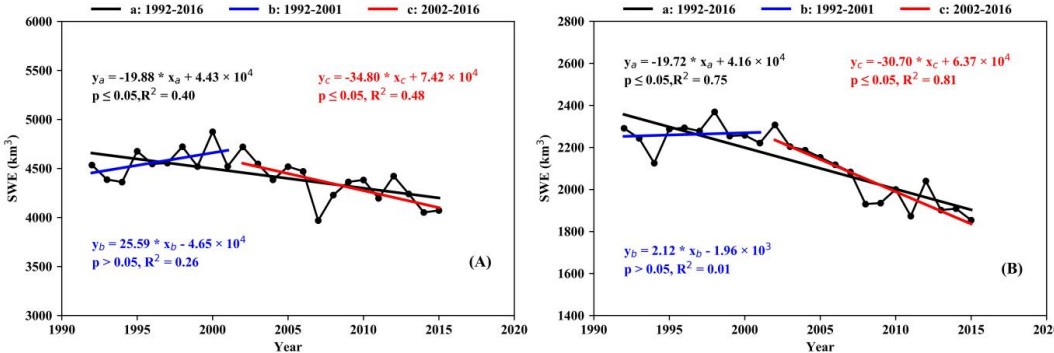

**Figure 10.** Interannual variation of annual maximum snow mass (A) and annual average snow mass (B) over Northern Hemisphere for three periods 1992-2016 (black line), 1992-2001 (blue line), and 2002-2016 (red line). Trends estimates were computed from least squares. P is the confidence level for the coefficient estimates; $R^2$ is the goodness of fit coefficient.

15 The snow mass variation characteristic over the past 25 years was explored by interannual variation (Fig. 10) and intra-annual cycles (figure not shown) of snow mass over the Northern Hemisphere. Fig. 10 depicts the time series of interannual variation of annual maximum and average snow mass for the 1992–2016 period. The largest value of annual maximum snow mass occurred in 1998–1999 up to ~ 4800 km³, while the least was ~ 4000 km³ in 2007-2008. The annual maximum snow mass presents a significant decreasing trend ($P \leq 0.05$) during the 1992–2016 period, at the rate of approximately -19.88 km³





yr.[-1] (Fig. 10A). Trend analysis reveals that annual maximum snow mass has undergone an 8% reduction from 1992 to 2016. Note that it presents an increase variation trend by about 25.59 km$^3$ yr.[-1] (P > 0.05) for 1992-2001. In contrast, the annual maximum snow mass exhibits a significantly decreasing trends (with -34.80 km$^3$ yr.[-1], P ≤ 0.05) since 2002 to 2016, which would lead to an extraordinary decrease during the 1992–2016 period. The greatest and the smallest value of annual average

snow mass respectively appear in 1998-1999 (~2370 km$^3$) and 2015-2016 (~1850 km$^3$) in Fig 10B. Likewise, in Fig 10B the annual average snow mass exhibit a significant decreasing trend during the 1992-2016 period at a rate of -19.72 km$^3$ yr.[-1] (P ≤ 0.05), and during the 2002-2016 period at a rate of -30.70 km$^3$ yr.[-1], P ≤ 0.05 (for 1992-2001, -2.2 km$^3$ yr.[-1]; P > 0.05). Furthermore, the annual average snow mass has undergone a 13% reduction during the 1992 – 2016 period. The factors, for instance, oceanic and atmospheric heat transport, sea ice, season wind, and solar insolation anomalies may contribute to the

fluctuation of snow mass (Liu and Key, 2014). Variation in snow mass across the Northern Hemisphere could well capture the variation characteristic of the Arctic sea ice extent (Tilling et al., 2015).

**Table 7.** Variation rate and changes of monthly average snow mass during 1992-2016. The asterisk indicate that the changes are significant at 95% confidence level. The changes were calculated with respect to the average of monthly average snow mass on 25 years.

| Month | Variation rate (km$^3$/yr.) | The percentage of Changes |
|---|---|---|
| September | -5.96* | -63.89% |
| October | -25.50* | -43.99% |
| November | -36.50* | -26.96% |
| December | -32.66* | -5.00% |
| January | -34.38* | -9.53% |
| February | -30.89* | -11.91% |
| March | 1.90 | -4.30% |
| April | -4.29 | -6.46% |
| May | -11.33* | -19.59% |
| June | -8.01* | -64.67% |

When analyzing long-term variation of monthly average snow mass (refer to Eq. B in Appendix), ten months (September

to June) exhibit significant decrease apart from March and April (Table 7). The maximum decrease rate was approximately -36.50 km$^3$ yr.[-1] (P ≤ 0.05) in November, while the minimum decrease occurred in April at -4.29 km$^3$ yr.[-1] (P > 0.05). Compared with the fall (September to November) and spring (March to June), the interannual variability of monthly average snow mass significantly decreased in winter (December to February), with an average rate of less than -32 km$^3$ yr.[-1]. We calculated the reduction of monthly average snow mass over 10 months with respect to the monthly average pattern over 1992-2016. The

reduction of monthly average snow mass fluctuated in a range from -65% to -4% for each month (September to June) over 1992-2016 (Table 7). The greatest and the smallest reduction are approximately 64.67% and 4.30%, which occurred in June

and March, respectively. Variation analysis of monthly average snow mass could offer powerful evidence of a significant decreasing trend for the annual average snow mass (Table 7, Fig. 10B).

Mudryk et al. (2015) compared five Northern Hemisphere SWE data sets and revealed a large amount of spread in total snow mass of these SWE datasets. We found that the average annual maximum snow mass of NHSnow products in the Northern Hemisphere are almost the same order of magnitude as multiple SWE datasets involved by Mudryk et al. (2015); the average annual maximum snow mass is approximately $4000 \pm 500$ km$^3$ for these datasets (excluding Greenland and Iceland). For the variation trend, the observation data and the simulated results by CMIP5 (Coupled Model Intercomparison Project phase 5) models exhibit a significant decreasing trend in the Northern Hemisphere. As same as the finding in this study (Fig. 10), the variation rate of decline becomes larger with entering the new century (after 2000) (Jeong et al., 2016). Although we analyzed the variation of snow mass, the analysis of the SWE in this study is still insufficient.

### 4.3 Snow cover days

Snow cover days (SCD) is defined as the number of days that SD is over 0 cm in a snow cover year (from September 1 through June 30) (Zhong, 2014). This study investigated the variation of SCD during 1992 – 2016. Most areas across the Northern Hemisphere present a prominent decreasing trend, with the rate ranging from 0 to 5 day yr.$^{-1}$ (Fig. 11a). The decreasing trend regions are mainly distributed in Eurasia, covering a vast area, e.g. the northern part of the Russia and large parts of central Asia. The regions with decrease rate greater than 5 day yr.$^{-1}$ are almost all located in China, such as the north of the Qilian Mountain, the central Tibetan Plateau, and Mount Tianshan. The regions with increasing trends can be found in the central North America, Western Europe, Northwestern Mongolia, and some areas of China. Throughout the Northern Hemisphere (Fig. 11b), the decreasing trend covers most areas of 25 through 85 °N with a mean decrease rate of approximately 1.0 day yr.$^{-1}$. The most notable variation (decrease or increase) occurs in polar regions (Fig. 11b). This may be because there are few pixels in the polar mainland regions (north of 70 °N).

Unlike SCD variation rate patterns, the variation trend patterns show that the non-significant change regions dominate SCD variation trends across the Northern Hemisphere (Fig. 11c). The extremely significant and the significant decreasing trend appear in the northwest of Hudson Bay in Canada, the Kamchatka peninsula, the Eastern European plains, the north of Russia, the Iranian plateau and, several regions in China (the Tibet Plateau, Mount Tianshan, and Northeast China Plain). In addition, the extremely significant and significant increase only occur in a limited area of North America and the eastern Qinghai–Tibetan Plateau region. Overall, the areas that show a significant decreasing trend are greater than that those that show a significant increasing trend.

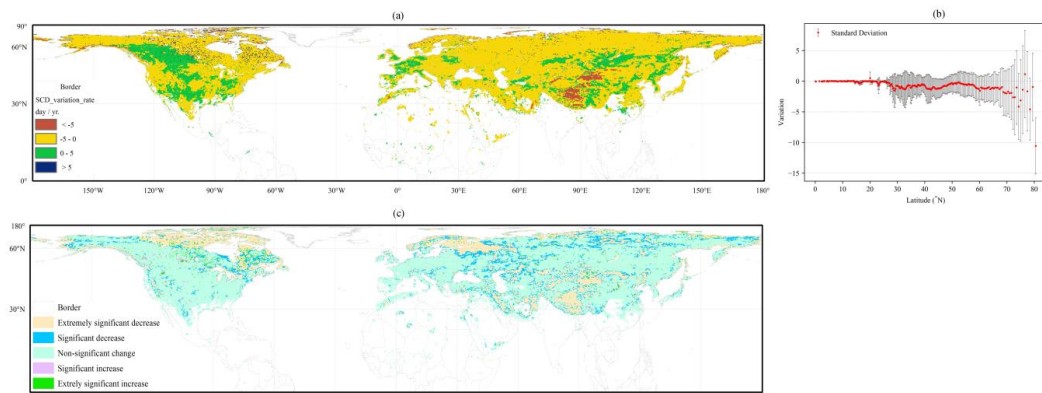

**Figure 11.** The variation rate pattern of SCD (a) and their statistical significances (c) over the Northern Hemisphere from 1992-2016. The zonal distribution in (b) is mapped at a 0.25-degree resolution in latitude. The error bars in (b) is one times of standard deviation.

Maximum SD in spring (Fig. 8c) and annual average SD (figure not shown) show the significant increasing trend, whereas SCD presents a significant decreasing trend in corresponding regions (Fig. 11c). Interestingly, this opposite trend of variation in SCD and SD appear in several regions, such as the regions in the plateau of Eastern Europe, the north of Central Siberian Plateau, the Greater Khingan Mountains of China, and the eastern Scandinavian Peninsula. Zhong et al. (2018) also reported this different variation trend of SD and SCD using ground-based measurement data over Eurasia. The primary reason for the increase of SD may be due to the increase in the frequency of extreme snowfall. Additionally, a recent study found that the greater SWE, the faster melting rate leading to a shortened SCD in the Northern Hemisphere (Wu et al., 2018).

However, when compared to SCD derived from optical sensor data, the specific quantity of SCD and SCD variation rate derived from NHSnow data was overestimated. Since the optical (MODIS or AVHRR) and microwave sensors (SSM/I or AMSR-E) respond in different parts of the electromagnetic spectrum, the estimated snow cover area and SCD from these two sensors would vary somewhat (Hori et al., 2017; Wang et al., 2018). The shallow snow could not induce volume scattering at 37 GHz, and thus passive microwave observations often provide better snow cover result at thick snow (> 5 cm) (Foster et al., 2009; Wang et al., 2008). The threshold for SCD definition here is 0 cm, whereas it is 1 cm or larger in other studies (Ke et al., 2016; Dyer and Mote, 2006). Notwithstanding, another explanation for these discrepancies may be the snow cover identification algorithm (Liu et al., 2018; Hall et al., 2002; Foster et al., 1997).

The microwave radiation characteristics of snow cover are similar to that of precipitation, cold desert, and frozen ground (Grody and Basist, 1996). Commission and omission errors in the NHSnow product may result from coarse spatial resolution, snow characteristics and topography (Dai et al., 2017), precipitation (Liu et al., 2018; Grody and Basist, 1996) especially over frozen ground (Tsutsui and Koike, 2012). Several rules for NHSnow algorithm development were applied to distinguish snow from precipitation, cold desert, and frozen ground (Xiao et al., 2018); it is impossible to entirely remove interference factors in each image. Moreover, the precondition of NHSnow is dry snow, which means almost no wet snow was considered inn the



SCD variation analysis (Singh and Gan, 2000). The poorer performance of the microwave derived products was anticipated because of documented difficulties in monitoring snow cover over forested and mountainous terrain (Vander Jagt et al., 2013; Smith and Bookhagen, 2016).

**5 Conclusions**

This study applied the SVR snow-depth retrieval algorithm developed by Xiao et al. (2018), which uses PM remote sensing and other auxiliary data, to generate long term (January 1, 1992, to December 31, 2016) Northern Hemisphere daily SD and SWE products (NHSnow) with 25-km spatial resolution. When compared to the other snow cover data sets (GlobSnow and ERA-Interim/Land) when benchmarked against the ground SD measurements, the NHSnow SD products had a relatively smaller bias (-0.59 cm), MAE (15.12 cm), and RMSE (20.11 cm). We then analyzed the spatial and temporal change in snow cover (SD, snow mass, and SCD) across the Northern Hemisphere, and quantified the magnitude of variation of snow cover

using SD and SWE extracted from NHSnow product. Our analysis results show that the SD variation pattern in variation rate or variation trend (increase or decrease) varies greatly among the different areas (Fig. 8 and 9). On the whole, the SD presented decreasing trends in three seasons (fall, winter, and spring) and in winter (from December to February) the variation rate is the highest up to $-0.11 \pm 0.40$ cm yr.$^{-1}$ (Table 6). From 1992 to 2016, the variation trends of annual maximum and average snow

mass also presented significant decrease trends at rate of -19.88 km$^3$ yr.$^{-1}$, -19.72 km$^3$ yr.$^{-1}$ (Fig. 10), respectively, and have undergone an 8% and 13% reduction, respectively. In almost every month (September to June), the monthly average snow mass exhibited a significant decreasing trend ($P \leq 0.05$), except for March and April. When seeing the SCD variation trend pattern, we found that most areas of the Northern Hemisphere showed a non-significant variation trend, and the spatial distribution pattern of SCD also is greatly different in different regions (Fig. 11).

While this study shed light on the spatiotemporal variability trends of snow cover across the Northern Hemisphere using the 25-year NHSnow product, we cannot claim that the NHSnow dataset could completely capture the climate change signal in each region and season. Because of the deficiencies and limitations (e.g. overestimation, underestimation), further efforts should be made to improve the estimation accuracy and robustness of the SD inversion algorithm. Additionally, when more reliable and numerous data become available, we will perform a more comprehensive validation over higher latitudes and

mountainous regions (Dai et al., 2017). Meanwhile, the validation analysis should also be carried out in complex terrain and different land cover types (Tennant et al., 2017; Snauffer et al., 2016). So far, further analyses and study are still need to help us to deeply understand the changes of SWE in the northern hemisphere, e.g. analyzing the difference of snow mass variation and its response to climate change in two major continents (Eurasia and North America) (Takala et al., 2011; Jeong et al., 2016) and investigating the variation trends of the peak of SWE in response to climate change in regional or hemispheric regions

(Irannezhad et al., 2016; Musselman et al., 2017; Brown and Mote, 2009; Zeng et al., 2018). It is recommended that future





work focus on the climatic effects and climatological causes in snow cover changes to comprehensively understand the associated snow cover change mechanisms against a climate change background (Huang et al., 2017; Flanner et al., 2011; Cohen et al., 2012; Mudryk et al., 2015).

*Competing interests*: We declare that we have no competing interests.

5     *Acknowledgments*: This study was funded by the National Natural Science Foundation of China (grant nos. 91325202; 41871050; 41801028), National Key Scientific Research Program of China (grant no. 2013CBA01802), and the Strategic Priority Research Program of Chinese Academy of Sciences (grant nos. XDA20100103; XDA20100313).

**Appendix**

$$SD_{average} = \frac{\sum_{i=1}^{n} SD_i}{n} \qquad (A)$$

$$SM_{average} = \frac{\sum_{i=1}^{n} SM_i}{n} \qquad (B)$$

Where $n$ is the number of days in one specific period of time (one month, or snow cover year/season), $i$ is $i$th day in one

10     specific period of time (one month, or snow cover year/season). SD is snow depth. SM is snow mass.

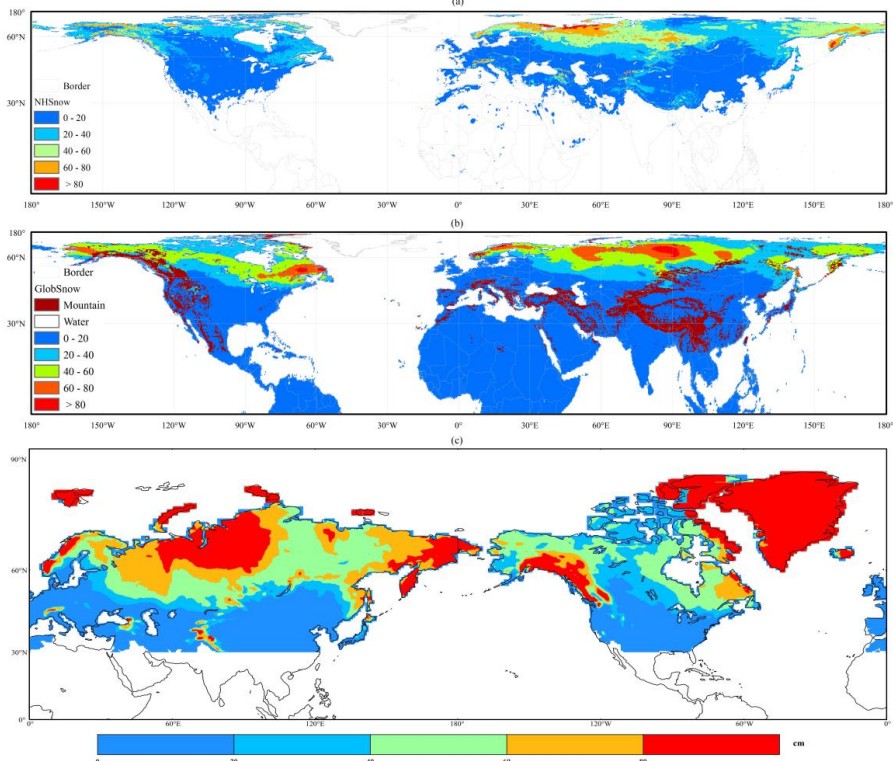

**Figure A**. Monthly average snow depth climatology of three products in February during 1992-2010: a) NHSnow; b) GlobSnow, c) ERA-Interim/Land



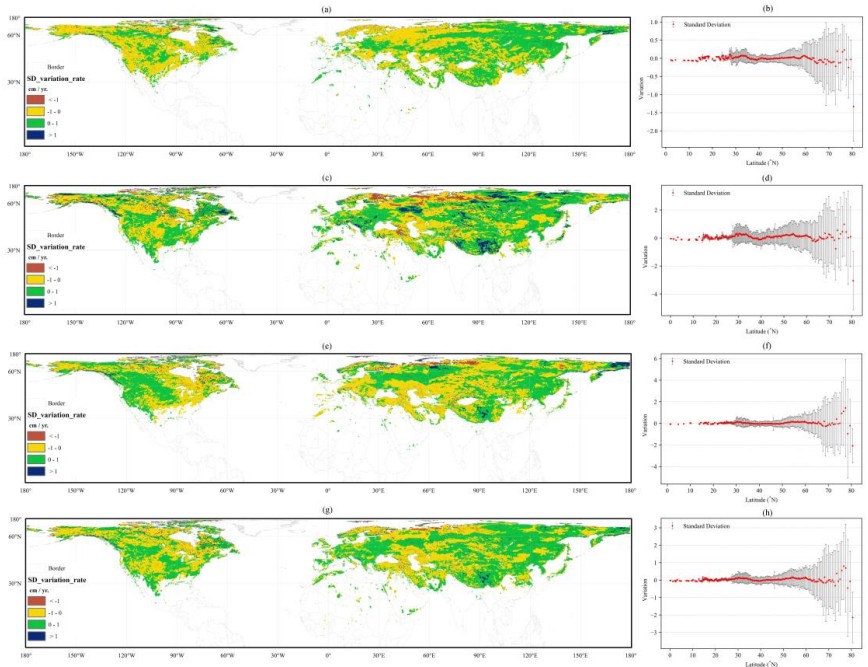

**Figure B.** The variation rate pattern of annual average (season) SD over the Northern Hemisphere for three snow cover season, fall (a, b; September to November), winter (c, d; December to February), spring (e, f; March to June) from 1992-2001. Black dots in (a, c, e, g) indicate that the changes are significant at 95% confidence level (CL). The zonal distribution in (b, d, f, h) are mapped at 0.25 degree resolution in latitude. The error bars in (b, d, f, h) is one times of standard deviation.

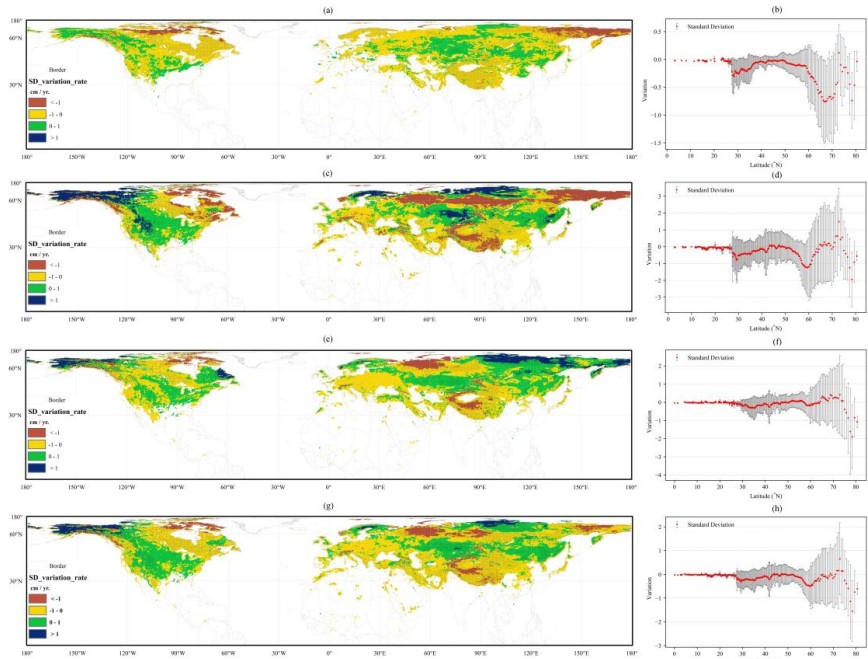

Figure C. The variation rate pattern of annual (season) average SD over the Northern Hemisphere for three snow cover season, fall (a, b;



September to November), winter (c, d; December to February), spring (e, f; March to June) from 2002-2016. Black dots in (a, c, e, g) indicate that the changes are significant at 95% confidence level (CL). The zonal distribution in (b, d, f, h) are mapped at 0.25 degree resolution in latitude. The error bars in (b, d, f, h) is one times of standard deviation.

5    **Table A.** AVHRR Global Land Cover classification and reclassification schemes

| Value | Classification Label | Reclassification Label |
|---|---|---|
| 0 | Water | Water |
| 1 | Evergreen needle leaf forest | Forest |
| 2 | Evergreen broad leaf forest | |
| 3 | Deciduous needle leaf forest | |
| 4 | Deciduous broad leaf forest | |
| 5 | Mixed forest | |
| 6 | Woodland | |
| 7 | Wooded grassland | Prairie (Grassland) |
| 10 | Grassland | |
| 8 | Closed shrub land | Shrub |
| 9 | Open shrub land | |
| 11 | Cropland | Bare-land |
| 12 | Bare ground | |
| 13 | Urban and built | |



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
