# Peer review of "Spatiotemporal variation of snow depth in the Northern Hemisphere from 1992 to 2016"

_The Cryosphere, 2019_

## Referee Comment (RC1) · Anonymous Referee #1 · 23 Jan 2020

General comments

The paper presents a new snow product, NHSnow, for SD and SWE derived using the support vector regression method. The product is validated by comparing it and other relevant SD products to in situ observations.

Validation of SWE product was omitted due to unavailability of SWE station observations. However, there is e.g. the ERA-CLIM2 data set available (http://litdb.fmi.fi/eraclim2.php). It contains snow course observations (SWE, density, SD) from Russia, Canada and Finland from 1935 to 2014. This does not cover the whole Northern Hemisphere or all of the time period considered in the paper, but provides good in situ data for validation. Please add some validation of the SWE product.

[Figure]

All map figures have too small fonts and are illegible. Please increase the font size and colorbars. Now over half of the maps shows regions without snow. Consider polar projection for the maps, this would emphasize the northern snow regions.

Specific comments

p1r18: Specify that the given error estimates are for SD.

p1r21: Specify that 13 % reduction in snow mass is from 1992 to 2016, not yearly.

p6r15-16: If I understand correctly, the data set provides SD and density, and you calculated SWE from these. Then "SWE, which is labelled as SD" is wrong, replace by "We calculated SWE from SD and density".

p6r18: "The time of this reanalysis data with the analysis type used in this study is to maximize the proximity to descending orbit time" is unclear. Please rephrase.

p7r9-14: List the snow properties and other parameters that you actually used, not examples. Also add a short description of SVR, at least I wasn't familiar with this method at all.

p7r20 and p8r6: How was the measured SD upscaled from station scale to satellite scale? How did you actually use measured SD?

p7r20: What does "indirectly considered" mean?

p10r12: Why did you set SD to 5 cm?

p14r5: Weather station information is very sparse in "the polar region and along the coast", and this is also probably the area with the deepest snow cover. Passive microwaves have a saturation point, so they cannot estimate deep snow accurately, and also the distance to in situ stations providing a priori information is greater than elsewhere. Please add some discussion on these points.

p14r13-14: From Figs 5-7 it seems like there are more red points (biggest error) in

[Figure]

NHSnow than in GlobSnow. Still you calculate larger bias and MAE for GlobSnow.

p14r16-17: Your product also uses measured SD. Is it different SD from the one used for validation? Now you claim that only GlobSnow uses the same data in assimilation than what was used in validation.

p15 r1-2: The GlobSnow algorithm and HUT model don't really use "evolution of snow grain size". The measured in situ snow depths are used to retrieve grain size, which is then used as input to the SWE retrieval, but grain size is varied within certain limits in the process. Therefore grain size is varying and not fixed, but I wouldn't call it evolution, as there is no physical snow model driving this change.

p15r10: Insensitivity to high SD is inherent in all algorithms based on attenuation of radiation in a media.

p15r19-21: You could consider fractional land cover, if you want to improve this. The spatial resolution of your land cover data is much better than the resolution of your PM data.

p18r5: If the rate of change is -0.11+/-0.40 cm/year, then your error estimate is so large that the trend could actually be positive. Please comment.

p18 Fig9 and p25 FigB and FigC: Where are the black dots in the figure?

p24 FigA: Use same map projection for all maps. Now only c) is different from all others in the paper. Also use the same color scales in all maps of same figure.

Technical corrections p2r11: "clouds have become" -> clouds are p2r22: SSM/S -> SSM/I p2r26: operated -> operate p4r11: equal-area scale earth -> Equal Area Scalable Earth p4r21: "which was created" -> begins? p7r14: us->is p15r2: Remove "was" p15r14: caused -> cause p18r19: 1996->2016 p19r13: Trends -> Trend p20r3: trends -> trend p20 Table 7: The percentage of Changes -> Percentage change p21r8: As same as -> Similar with p21r27: "than that those that" -> than those that p23r16: "undergone an 8% and 13% reduction" from 1992 to 2016.

---

## Referee Comment (RC2) · Anonymous Referee #2 · 7 Feb 2020

This paper presents a Northern Hemisphere daily snow depth (SD) and snow water equivalent (SWE) product (NHSnow) over the 1992-2016 period, by applying a support vector regression snow depth retrieval algorithm, already published by the same team (Xiao et al., 2018, RSE). This algorithm uses passive microwave (PM) remote sensing (RS) data (SSM/I and SSMIS) and auxiliary data such as in-situ meteorological and snow depth data for training, and an empirical snow density model for SWE retrieval. Only dry snow is considered in this retrieval since it is based on PM data. Performances of this NHsnow dataset against in-situ SD data was compared to those of Globsnow2 (GB) and ERA-Interim reanalysis (ERAi). SWE retrievals were not evaluated. Results show that NHsnow SD is of the same order of magnitude than GB and ERAi for bias, mean absolute error (MAE) and RMSE, expected a slight smaller mean bias of 0.59

cm, compared to – 1.19 cm (GB) and 5.6 cm (ERAi). Even if the method used appears interesting (presented in another paper already published), I don't see the real added-value of this dataset? The methods remain dependent on in-situ observations (needed for training), these in-situ data are sometimes sparsely distributed particularly in the North, giving point measurements against 25 km resolution... The known limitations from using PM data (wet snow, deep snow, mountainous area...) are not discussed, and seem not improved? Furthermore, the SWE retrieval is based on an empirical density equation that leads to non-validated SWE values! Thus, the motivations for using such dataset remains unclear given the numerous other databases?

Moreover, the literature review presented for SD and SWE retrievals is incomplete. The authors ignore recent results from assimilation of RS data in Land Surface Model, including improved snow model, driven by meteorological data (and/or reanalysis). Such approaches are more interesting given their independent from in-situ snow measurements and provide both SD and SWE data (See Larue et al., 2018, Hydrol. Earth Syst. Sci., 22; Kwon et al. 2016, J. Hy- drometeorol., 17, 2853–2874; Charrois et al., 2016, The Cryosphere, 10:1021–1038; De Lannoy et al., 2012, Water Resour. Res., 48, W01522). Also recent active PM SAR-based analysis can provide SD data at high spatial resolution : coherence analysis (Singh et al., Water 2020, 12, 21) or phase difference from ESA Sentinel constellation, Leinss, S.; Parrella, G.; Hajnsek, I. Snow height determination by polarimetric phase differences in X-band SAR data. IEEE J. Sel. Top. Appl. Earth Observ. Remote Sens. 2014, 7, 3794–3810), also completely independently from in-situ data!

In their paper, the authors analyzed also the trend of SD (mean and max), SWE, Snow Cover Extent (SCE) and Snow Cover Duration (SCD), showing similar known results than those already published. There are no really new insights here, even if the results are well presented with maps showing spatial variability between North Hemisphere regions (excepted trends slighted over too short periods, see bellow). Also, the authors do not discuss the fact that results based on dry snow only are biased in spring when

snow is generally wet. Finally, this paper brings any explanation on the observed trends (some period and areas with increase or decrease snow parameters), as the authors recognized at the end of the paper.

Overall, I recognize that to produce a global dataset is a strong work and that the authors succeed to reach the mean accuracy level of existing databases, but this paper is relatively weak in its original scientific contribution (any real improvement; trends more or less known). I thus don't recommend its publication in TC.

This paper describing the NHsnow database should be submitted to the dedicated journal for new released datasets: Earth Syst. Sci. Data.

Specific comments 1. Introduction: incomplete literature review about other approaches. Also, limitations of SWE retrieval based on PM are not well reviewed. One of the main problem is the snow microstructure (grain size, stratigraphy, ice crust layer...) that evolves during the winter and that strongly affects the PM emission, more than SWE! (see Sandells et al.,2017, The Cryosphere, 11, 229–246; Roy et al., 2016, The Cryosphere, 10; Durand et al., 2011, IEEE Geosci. Remote Se., 8 ; ... and Matzler, 1987, Remote Sens. Rev., 2, 259–387).

3.3 Estimation of SWE Very empirical approach (Eq. 3 and Table 3), and without statistical error analysis? PM data are known to be limited over deep snow (see Larue et al., 2017, Remote Sens. Environ., 194).

4. Results Yes, in-situ SWE datasets exist for data over Siberia (Bulygina, O., Groisman, P. Y., Razuvaev, V., and Korshunova, N. (2011). Changes in snow cover characteristics over northern eurasia since 1966. Environmental Research Letters, 6(4):045204) and over Canada (Brown, R. D., Fang, B., and Mudryk, L. (2019). Update of canadian historical snow survey data and analysis of snow water equivalent trends, 1967-2016: Research note. Atmosphere-Ocean, 1-8).

All the maps are too small, hard to read. Seasonal trend analysis biased when based

on dry snow. Have you eliminated wet snow from ERAi outputs?

4.2 Snow mass trend I don't agree with the snow mass trend over too short periods (1992-2001) and 2002-2016) (Fig. 10). A trend over only 10 years makes no sense: you only change one value in the series, and the slope changes drastically! Such analysis has no interest here (maybe for sensationalism public journals!) Analysis of SWE is insufficient.

4.3 Snow cover days: the usually term used is "Snow Cover Duration" (SCD)

5. Conclusion No convincing arguments for using NHsnow instead of others? (added value?, improvements?).

---

## Author Comment (AC1) · 19 Apr 2020

**Response to Reviewer Comments**

We would very much like to thank the editor and reviewer for the constructive and insightful comments on this manuscript. We have carefully revised the manuscript and provided point-by-point response to each of the comments below. The comments are in black and our responses are in blue (the revised sentence was set in *italics*). For each comment we have indicated how we have changed the manuscript to address the comments in the revised version.
* * *
********************Reply to comments from the anonymous reviewer 1#********************
* * *
**REVIEWER 1#**

General comments

The paper presents a new snow product, NHSnow, for SD and SWE derived using the support vector regression method. The product is validated by comparing it and other relevant SD products to in situ observations.

Validation of SWE product was omitted due to unavailability of SWE station observations. However, there is e.g. the ERA-CLIM2 data set available (http://litdb.fmi.fi/eraclim2.php). It contains snow course observations (SWE, density, SD) from Russia, Canada and Finland from 1935 to 2014. This does not cover the whole Northern Hemisphere or all of the time period considered in the paper, but provides good in situ data for validation. Please add some validation of the SWE product. All map figures have too small fonts and are illegible. Please increase the font size and colorbars. Now over half of the maps shows regions without snow. Consider polar projection for the maps, this would emphasize the northern snow regions.

Response: Thank you very much for your constructive suggestions and comments. The images in the revised manuscript were updated. Additionally, we downloaded the SWE in-situ observations from the website provided. The revised version of the manuscript were added the evaluation and validation of the SWE for NHSnow and GlobSnow products in Section 4.1.2, as following:

*"To perform the SWE evaluation, we acquired more than 77 000 valid data records of NHSnow, GlobSnow, and in-situ measurements from 1992 to 2014 (December to February). We conducted performance for both NHSnow and GlobSnow SWE products in shallow (< 150 mm) and deep (≥ 150 mm) snow conditions. (Larue et al., 2017). The performance metrics were summarized in Table 5 for both NHSnow and GlobSnow SWE products against snow course observation over the former Soviet Union, Canada, and Finland. The overall bias, MAE and RMSE for NHSnow SWE products are 43.6 mm, 61.9 mm and 87.3 mm respectively; while for GlobSnow, they are 15.3 mm, 31.6 mm, 61.5 mm respectively. For shallow snow condition (SWE < 150 mm), the bias, MAE and RMSE are slightly reduced for both SWE products compared to using total records condition (Table 5). Nevertheless, when analyzing the deep SWE (≥ 150 mm), the statistics results for both SWE products are relative large (NHSnow: bias = -46.0 mm, MAE = 103.6 mm, RMSE = 169.0 mm; GlobSnow: bias = -103.1 mm, MAE =*

*112.2 mm, RMSE = 201.0 mm; Table 5). In general, the GlobSnow SWE products have a better performance than NHSnow SWE product, especially in shallow snow condition (< 150 mm). However, for deep snow, NHSnow SWE product has a better performance, with less bias, MAE, and RMSE, than GlobSnow SWE product.*

*The evaluation performance of both NHSnow and GlobSnow SWE products was conducted with respect to SWE in-situ observations over the former Soviet Union, Canada, and Finland (Fig. 9). For GlobSnow SWE product, the SWE estimation have the best performance in the former Soviet Union region area than the other two regions (Canada and Finland) with the least bias and MAE; For GlobSnow SWE products, the estimation performance in Finland region which with less bias, while MAE and RMSE is better than that in Canadian regions (Fig. 9). Our analysis result is consistent with the accuracy evaluation results from previous published study in Canadian regions (Larue et al., 2017). Through analyzing the SWE in-situ observations, we found that omission error of snow cover identification, which means that in-situ observation is fully snow-covered while the prediction of snow cover algorithm is snow-free, is the main error source for GlobSnow SWE product. The omission error of GlobSnow SWE product in the former Soviet Union, Canada, and Finland are 6.1%, 18.4 and 8.2% respectively. As for NHSnow SWE product, the best estimation performance is in Canada; then in the former Soviet Union; the estimated SWE in Finland was not as good as the regions (Fig. 9). Unlike GlobSnow, the NHSnow SWE products do not have the omission error but the commission error, which is defined as snow-free observed by in-situ data but snow-covered detected by snow cover algorithm. The commission errors for NHSnow product are 0, 8.9%, and 0 for the former Soviet Union, Canada, and Finland respectively. Though the commission error is 8.9% in Canada, the NHSnow SWE estimation performance (less bias and MAE; -7.9 and 48.9) is better than in the other two regions. It may be due to the statistical model of snow density, which is obtained through the snow observation data across North America (Sturm et al., 2010). In other words, this snow density model has a better applicability in North America (Hill et al., 2019), but may not be the case in the other two regions. The large RMSE in Canada than in the former Soviet Union may be due to deep snow because there is 10.4% of deep snow in Canada. Snow density may also contribute to misestimating SWE. For NHSnow SWE product in the former Soviet Union region with 6.1% deep snow records, there is no commission error and the error of the estimated snow depth is relatively small (bias < 5 cm, MAE < 15 cm from Section 4.1.1), we thus assume that the errors in SWE primarily come from the modeled snow density in the former Soviet Union region. Through analyzing SWE observation and SD estimates of NHSnow, the performance in Finland may be accounted for by two most possible reasons which are deep snow (more than 8% records is deep snow) and the inaccurate estimates of SD (bias = -13.3, MAE = 18.3 from Section 4.1.1). Additionally, there are no commission error in Finland region; therefore, the inaccurate SD estimates would be the main error source of NHSnow SWE product in Finland region.*

*Based on the above analysis, NHSnow does a fairly good work in SD estimation (bias = -0.59 cm, MAE = 15.98 cm, and RMSE = 20.11 cm). Although deep snow is a great challenge for current SWE products, NHSnow SWE product have less error in deep snow compared to GlobSnow SWE product (Table 5, Fig. 9). Moreover, the statistical snow density model proposed by Sturm et al. (2010) cannot accurately describe the evolution of snow density in Eurasia region, and it may be a*

*major source of error for NHSnow SWE product, which needs further investigation when related data are available.*

*Table 5. Summary of performance indexes (Bias, MAE, RMSE; unit: mm) for NHSnow SWE product and GlobSnow SWE product against SWE in-situ measurements during 1992 to 2014 (December–February)*

|  | Total | | < 150 mm | | ≥ 150 mm | |
|---|---|---|---|---|---|---|
|  | *GlobSnow* | *NHSnow* | *GlobSnow* | *NHSnow* | *GlobSnow* | *NHSnow* |
| *Bias* | *15.3* | *43.6* | *11.0* | *41.4* | *-103.1* | *-46.0* |
| *MAE* | *31.6* | *61.9* | *25.6* | *59.9* | *112.2* | *103.6* |
| *RMSE* | *61.5* | *87.3* | *32.8* | *77.8* | *201.0* | *169.7* |

[Figure]

*Figure 9. The performance evaluation of two SWE products (NHSnow and GlobSnow) with respect to SWE in-situ observations over the former Soviet Union, Canada, and Finland using three indexes (Bias, MAE, RMSE; unit: mm)*
*"*

Specific comments

p1r18: Specify that the given error estimates are for SD.

Response: Thank you. I modified the original sentence "The evaluation results show that NHSnow performs comparably well with relatively high accuracy (bias: -0.59 cm, mean absolute error: 15.12 cm, and root mean square error: 20.11 cm) when benchmarked against the station snow depth measurements" to "*The evaluation results show that NHSnow dataset performs comparably well with relatively high accuracy for snow depth with bias of -0.59 cm, mean absolute error of 15.98 cm, RMSE and root mean square error of 20.11 cm when benchmarked against the station snow depth measurements*" in Abstract.

p1r21: Specify that 13 % reduction in snow mass is from 1992 to 2016, not yearly.

Response: Thanks. We changed it to "…, *and subjects to about 13% reduction in snow mass from 1992 to 2016*" in Abstract.

p6r15-16: If I understand correctly, the data set provides SD and density, and you calculated SWE from these. Then "SWE, which is labelled as SD" is wrong, replace by "We calculated SWE from SD and density".

Response: Thank you. The sentence "SWE, which is labelled as the SD of water equivalent in this dataset, is one of the thirteen parameters provided" was rephrased to "*SWE with unit in m of water equivalent is one of the thirteen parameters provided*" in page 7 line 19.

p6r18: "The time of this reanalysis data with the analysis type used in this study is to maximize the proximity to descending orbit time" is unclear. Please rephrase.

Response: This sentence is changed to "*In data pair extraction process, we make the data pair have a close observation/measurement time.*", and it was remove to page 7 line 5.

p7r9-14: List the snow properties and other parameters that you actually used, not examples. Also add a short description of SVR, at least I wasn't familiar with this method at all.

Response: We agree. According to your suggestion, it should illustrate the snow properties or parameters used in our study. In the paragraph 1 in Section 3.2, it have the similar description, "*Following Eq. 1, we used ten variables as the inputs, including PM brightness temperature (19 GHz, 37 GHz, 85 or 91 GHz) with vertical and horizontal polarizations, geographical location (latitude and longitude), elevation, and the measured SD.*"

In Section 3.2, we revised and added the description about SVR in page 8 lines 17-20.

"*Xiao et al. (2018) developed the SVR SD retrieval algorithm, which used a non-linear regression method (support vector regression, SVR) method as the retrieval function (g (·) in Eq. 1). This machine learning method called SVR (Üstün et al., 2005) has been applied in various fields, for instance, biology and medicine (Zhang and Ge, 2013), to establish the complex relationship between target variable and input variables.*"

Reference:

Zhang, G., and Ge, H.: Support vector machine with a Pearson VII function kernel for discriminating halophilic and non-halophilic proteins, Computational biology and chemistry, 46, 16-22, 2013.

p7r20 and p8r6: How was the measured SD upscaled from station scale to satellite scale? How did you actually use measured SD?

Response: As previous study general operation, the point-level measured SD directly matched the pixel-level SD value, and

we do not do further process. In training stage, the measured SD was input the training model as a true observation to build the snow depth retrieval model. For example, in the formula $y=ax + b$, $x$ (the snow properties related input variables) and $y$ (the measured SD) was first used to obtain the coefficient $a$ and $b$. In validation stage, the measured SD was used to evaluate the estimated results.

p7r20: What does "indirectly considered" mean?

Response: The "indirectly considered" means that the information on properties variation of snow with season was not used as an input variable of retrieval model. Conversely, the information on seasonal variation of snow cover properties was transformed to different snow cover stages (the snow cover season is divided into three stages: snow accumulation stage, stabilization stage, and ablation stage), which was basis of establishing snow depth retrieval models.

p10r12: Why did you set SD to 5 cm?

Response: The setting of this threshold is according to previous two studies (Che et al, 2016; Wang et al., 2008). This threshold will be optimized using more observational data in our following improved version of the algorithm.

Che, T., et al. : Estimation of snow depth from passive microwave brightness temperature data in forest regions of northeast China, Remote Sensing of Environment, 183, 334-349, 2016.

Wang, X., et al.: Evaluation of MODIS snow cover and cloud mask and its application in Northern Xinjiang, China, Remote Sensing of Environment, 112, 1497-1513, 10.1016/j.rse.2007.05.016, 2008.

p14r5: Weather station information is very sparse in "the polar region and along the coast", and this is also probably the area with the deepest snow cover. Passive microwaves have a saturation point, so they cannot estimate deep snow accurately, and also the distance to in situ stations providing a priori information is greater than elsewhere. Please add some discussion on these points.

Response: Thank you. We add the discussion about the deep snow and saturation effect at specific frequency. We add the following description information in page 19 lines 7-11.

"*In addition to above discussed factors, deep snow is another critical influence factor in SD and SWE retrieval. Because of the saturation of the penetration depth at 37 GHz, many studies indicate that deep snow is a major source of uncertainties in SD and SWE retrieval when using passive microwave brightness temperature (Roy et al., 2016;Durand et al., 2011;Larue et al., 2017;Saberi et al., 2019).*"

p14r13-14: From Figs 5-7 it seems like there are more red points (biggest error) in NHSnow than in GlobSnow. Still you calculate larger bias and MAE for GlobSnow.

Response: Thank you very much. We modified the values of MAE for GlobSnow and NHSnow in Table 4. We re-calculated the whole statistic indexes in Table 4 and show that the values of bias are consistent with the results in previous version manuscript. And also, the numbers in histograms of Figure 5 that were calculated by Python program indicates the GlobSnow have larger bias than NHSnow.

p14r16-17: Your product also uses measured SD. Is it different SD from the one used for validation? Now you claim that only GlobSnow uses the same data in assimilation than what was used in validation.

Response: Thanks for your comment. In data process, the measured SD datasets used for training model with special mark are different from the measured SD used for validation. We added the description "*Note that the data used for the validation is completely independent of the data generate the SD retrieval models*" in page 13 lines 10-11

p15 r1-2: The GlobSnow algorithm and HUT model don't really use "evolution of snow grain size". The measured in situ snow depths are used to retrieve grain size, which is then used as input to the SWE retrieval, but grain size is varied within certain limits in the process. Therefore grain size is varying and not fixed, but I wouldn't call it evolution, as there is no physical snow model driving this change.

Response: Thank you very much. We rephrased the sentence "The different performance for these two products may be mainly caused by the evolution of snow grain size was used by HUT (The Helsinki University of Technology) model to generate SWE in GlobSnow" to "*The different performance for these two products may be mainly caused by the variation of snow grain size which was used to generate SWE of GlobSnow*" in page 16 lines 3-4.

p15r10: Insensitivity to high SD is inherent in all algorithms based on attenuation of radiation in a media.

Response: Thanks. The original sentence was changed to "*..., they all struggle to capture SD with low bias, MAE, and RMSE in high SD accumulation (deep snow) regions (Fig. 6-8, Fig. A). In fact, all SD retrieval algorithms based on attenuation of radiation are insensitivity to deep snow*" in page 16 lines 13-14.

p15r19-21: You could consider fractional land cover, if you want to improve this. The spatial resolution of your land cover data is much better than the resolution of your PM data.

Response: Thank you very much for your suggestion. The improvement work of this algorithm is going on, and we will take account of fractional land cover data in following version algorithm.

p18r5: If the rate of change is -0.11+/-0.40 cm/year, then your error estimate is so large that the trend could actually be positive. Please comment.

Response: Thank you. The strong variability for the winter snow depth (great deviation) mainly due to regional variability

which have large the absolute value of variation rate. The winter snow depth especially in the polar region exhibit a significant increase trend. We added the comment in page 21 lines 14-17:

"*The snow depth in some polar regions was significant change (Fig. 11b), and the absolute values of the variation rate in these regions are apparently greater than in the middle-low latitude (Fig. 11; Figure D in the Appendix). These results are indeed consistent with station observations over northern of Russia (Bulygina et al., 2011), northern Canada (Brown et al., 2019) and Alaska. Therefore, the seasonal average SD in the Northern Hemisphere subjects to a strong variability (standard deviation).*"

Reference:

Ross D. Brown, Bruno Fang & Lawrence Mudryk (2019) Update of Canadian Historical Snow Survey Data and Analysis of Snow Water Equivalent Trends, 1967–2016, Atmosphere-Ocean, 57:2, 149-156, DOI: 10.1080/07055900.2019.1598843

Bulygina, O., Groisman, P. Y., Razuvaev, V., and Korshunova, N. (2011). Changes in snow cover characteristics over northern eurasia since 1966. Environmental Research Letters, 6(4):045204

p18 Fig9 and p25 FigB and FigC: Where are the black dots in the figure?

Response: Thanks. The black dots in the figure are not apparent because the dots in each pixel are so small. In the figure it show that the shade is in the color map. These figures (Fig 11, Fig B and C) were updated.

p24 FigA: Use same map projection for all maps. Now only c) is different from all others in the paper. Also use the same color scales in all maps of same figure.

Response: Thank you. The projection of Fig A-c was revised. Color scales are also updated.

Technical corrections

p2r11: "clouds have become" -> clouds are

Response: we changed "clouds have become the greatest hurdle…" to "*clouds are the greatest hurdle*" in page 2, line 13.

p2r22: SSM/S - > SSM/I

Response: we removed the repetitive description "Special Sensor Microwave Imager (SSM/S)" in page 2, line 24.

p2r26: operated -> operate

Response: we changed "operated" to "*operate*" in page 2, line 32

p4r11: equal-area scale earth -> Equal Area Scalable Earth

Response: The "equal-area scale earth" was changed to "*Equal Area Scalable Earth*" in page 5, line 3.

p4r21: "which was created" -> begins?

Response: "This online dataset, which was created in 1929, is …" was revised to "*This online dataset beginning 1929,*" in page 5 line 13

p7r14: us->is

Response: "us" was changed to "is" in page 8, line 14.

p15r2: Remove "was"

Response: "snow grain size was used …" was changed to "snow grain size used" in page 16, line 4.

p15r14: caused -> cause

Response: The "caused" was changed to "cause" in page 19 line 1.

p18r19: 1996->2016

Response: "2002-1996" in the title of Table 6 was revised to "2002-2016" in page 22, line 7.

p19r13: Trends -> Trend

Response: We changed the word "Trends" in title Figure 10 to "Trend" in page 23, line 9.

p20r3: trends -> trend

Response: "… exhibits a significantly decreasing trends" was changed to "… exhibits a significantly decreasing trend" in page 23, line 18.

p20 Table 7: The percentage of Changes -> Percentage change

Response: "The percentage of Changes" was changed to "*Percentage change*" in Table 8.

p21r8: As same as -> Similar with

Response: "As same as" was revised to "*Similar with*" in page 24 line 19.

p21r27: "than that those that" -> than those that

Response: we removed the second "that" in this sentence, in page 25, line 18

p23r16: "undergone an 8% and 13% reduction" from 1992 to 2016.

Response: we revised original description to "*undergone an 8% and 13% reduction from 1992 to 2016*" in page 27, line 17.

---

## Author Comment (AC2) · 19 Apr 2020

**Response to Reviewer Comments**

We would very much like to thank the editor and reviewer for the constructive and insightful comments on this manuscript. We have carefully revised the manuscript and provided point-by-point response to each of the comments below. The comments are in black and our responses are in blue (the revised sentence was set in *italics*). For each comment we have indicated how we have changed the manuscript to address the comments in the revised version.
* * *
******************* **Reply to comments from the anonymous reviewer 2#**********************
* * *
**REVIEWER 2#**

This paper presents a Northern Hemisphere daily snow depth (SD) and snow water equivalent (SWE) product (NHSnow) over the 1992-2016 period, by applying a support vector regression snow depth retrieval algorithm, already published by the same team (Xiao et al., 2018, RSE). This algorithm uses passive microwave (PM) remote sensing (RS) data (SSM/I and SSMIS) and auxiliary data such as in-situ meteorological and snow depth data for training, and an empirical snow density model for SWE retrieval. Only dry snow is considered in this retrieval since it is based on PM data. Performances of this NHsnow dataset against in-situ SD data was compared to those of Globsnow2 (GB) and ERA-Interim reanalysis (ERAi). SWE retrievals were not evaluated. Results show that NHsnow SD is of the same order of magnitude than GB and ERAi for bias, mean absolute error (MAE) and RMSE, expected a slight smaller mean bias of 0.59 cm, compared to – 1.19 cm (GB) and 5.6 cm (ERAi). Even if the method used appears interesting (presented in another paper already published), I don't see the real added value of this dataset? The methods remain dependent on in-situ observations (needed for training), these in-situ data are sometimes sparsely distributed particularly in the North, giving point measurements against 25 km resolution: : : The known limitations from using PM data (wet snow, deep snow, mountainous area: : :) are not discussed, and seem not improved? Furthermore, the SWE retrieval is based on an empirical density equation that leads to non-validated SWE values! Thus, the motivations for using such dataset remains unclear given the numerous other databases?

Response: Thanks for your comments and suggestions. Snow depth retrieval algorithm has been published in our previous work. However, our last work (Xiao et al., 2018) mainly concentrate on algorithm development, while the current manuscript is more focused on the data products and hemispheric scale analysis. In terms of specific difference, they are illustrated in Section 3.2 step by step. Overall, we briefly described the unchanged steps (steps 1, 2, 4 and 5) for the SVR SD retrieval algorithm and provided more detailed information for the changed steps (steps 3 and 6) and two new additional steps (steps 7 and 8) for the algorithm improvement.

As to SWE evaluation, the snow course observations in the Northern Hemisphere provided by Reviewer 1# were used to

validated and evaluated the SWE products (NHSnow and GlobSnow) in Section 4.1.2.

*"To perform the SWE evaluation, we acquired more than 77 000 valid data records of NHSnow, GlobSnow, and in-situ measurements from 1992 to 2014 (December to February). We conducted performance for both NHSnow and GlobSnow SWE products in shallow (< 150 mm) and deep (≥ 150 mm) snow conditions. (Larue et al., 2017). The performance metrics were summarized in Table 5 for both NHSnow and GlobSnow SWE products against snow course observation over the former Soviet Union, Canada, and Finland. The overall bias, MAE and RMSE for NHSnow SWE products are 43.6 mm, 61.9 mm and 87.3 mm respectively; while for GlobSnow, they are 15.3 mm, 31.6 mm, 61.5 mm respectively. For shallow snow condition (SWE < 150 mm), the bias, MAE and RMSE are slightly reduced for both SWE products compared to using total records condition (Table 5). Nevertheless, when analyzing the deep SWE (≥ 150 mm), the statistics results for both SWE products are relative large (NHSnow: bias = -46.0 mm, MAE = 103.6 mm, RMSE = 169.0 mm; GlobSnow: bias = -103.1 mm, MAE = 112.2 mm, RMSE = 201.0 mm; Table 5). In general, the GlobSnow SWE products have a better performance than NHSnow SWE product, especially in shallow snow condition (< 150 mm). However, for deep snow, NHSnow SWE product has a better performance, with less bias, MAE, and RMSE, than GlobSnow SWE product.*

*The evaluation performance of both NHSnow and GlobSnow SWE products was conducted with respect to SWE in-situ observations over the former Soviet Union, Canada, and Finland (Fig. 9). For GlobSnow SWE product, the SWE estimation have the best performance in the former Soviet Union region area than the other two regions (Canada and Finland) with the least bias and MAE; For GlobSnow SWE products, the estimation performance in Finland region which with less bias, while MAE and RMSE is better than that in Canadian regions (Fig. 9). Our analysis result is consistent with the accuracy evaluation results from previous published study in Canadian regions (Larue et al., 2017). Through analyzing the SWE in-situ observations, we found that omission error of snow cover identification, which means that in-situ observation is fully snow-covered while the prediction of snow cover algorithm is snow-free, is the main error source for GlobSnow SWE product. The omission error of GlobSnow SWE product in the former Soviet Union, Canada, and Finland are 6.1%, 18.4 and 8.2% respectively. As for NHSnow SWE product, the best estimation performance is in Canada; then in the former Soviet Union; the estimated SWE in Finland was not as good as the regions (Fig. 9). Unlike GlobSnow, the NHSnow SWE products do not have the omission error but the commission error, which is defined as snow-free observed by in-situ data but snow-covered detected by snow cover algorithm. The commission errors for NHSnow product are 0, 8.9%, and 0 for the former Soviet Union, Canada, and Finland respectively. Though the commission error is 8.9% in Canada, the NHSnow SWE estimation performance (less bias and MAE; -7.9 and 48.9) is better than in the other two regions. It may be due to the statistical model of snow density, which is obtained through the snow observation data across North America (Sturm et al., 2010). In other words, this snow density model has a better applicability in North America (Hill et al., 2019), but may not be the case in the other two regions. The large RMSE in Canada than in the former Soviet Union may be due to deep snow because there is 10.4% of deep snow in Canada. Snow density may also contribute to misestimating SWE. For NHSnow SWE product in the former Soviet Union region with 6.1% deep snow records, there is no commission error and the error of the estimated snow*

*depth is relatively small (bias < 5 cm, MAE < 15 cm from Section 4.1.1), we thus assume that the errors in SWE primarily come from the modeled snow density in the former Soviet Union region. Through analyzing SWE observation and SD estimates of NHSnow, the performance in Finland may be accounted for by two most possible reasons which are deep snow (more than 8% records is deep snow) and the inaccurate estimates of SD (bias = -13.3, MAE = 18.3 from Section 4.1.1). Additionally, there are no commission error in Finland region; therefore, the inaccurate SD estimates would be the main error source of NHSnow SWE product in Finland region.*

*Based on the above analysis, NHSnow does a fairly good work in SD estimation (bias = -0.59 cm, MAE = 15.98 cm, and RMSE = 20.11 cm). Although deep snow is a great challenge for current SWE products, NHSnow SWE product have less error in deep snow compared to GlobSnow SWE product (Table 5, Fig. 9). Moreover, the statistical snow density model proposed by Sturm et al. (2010) cannot accurately describe the evolution of snow density in Eurasia region, and it may be a major source of error for NHSnow SWE product, which needs further investigation when related data are available.*

*Table 5. Summary of performance indexes (Bias, MAE, RMSE; unit: mm) for NHSnow SWE product and GlobSnow SWE product against SWE in-situ measurements during 1992 to 2014 (December–February)*

| | *Total* | | *< 150 mm* | | *≥ 150 mm* | |
|---|---|---|---|---|---|---|
| | *GlobSnow* | *NHSnow* | *GlobSnow* | *NHSnow* | *GlobSnow* | *NHSnow* |
| *Bias* | *15.3* | *43.6* | *11.0* | *41.4* | *-103.1* | *-46.0* |
| *MAE* | *31.6* | *61.9* | *25.6* | *59.9* | *112.2* | *103.6* |
| *RMSE* | *61.5* | *87.3* | *32.8* | *77.8* | *201.0* | *169.7* |

[Figure]

*Figure 9. The performance evaluation of two SWE products (NHSnow and GlobSnow) with respect to SWE in-situ observations over the former Soviet Union, Canada, and Finland using three indexes (Bias, MAE, RMSE; unit: mm)*

,,

In term of deep snow and wet snow, we added the discussion about these two factors in Section 4 (Results and Discussion)

*"In addition to above discussed factors, deep snow is another critical influence factor in SD and SWE retrieval. Because of the saturation of the penetration depth at 37 GHz, many studies indicate that deep snow is a major source of uncertainties in SD and SWE retrieval when using passive microwave brightness temperature (Roy et al., 2016;Durand et al., 2011;Larue et al., 2017;Saberi et al., 2019)."* In page 19, lines 10-13.

*"We have to note that snow-covered area of NHSnow product was dry snow determined by Grody's series of rules (Grody and Basist, 1996). The snowpack inevitably contains liquid water since snow melting events occur especially in spring and summer. During spring and summer periods, consequently, the snow mass and SCD derived from NHSnow product will be partly underestimated and the bias may occur in the change trend analysis of snow cover"* in page 21, lines 23-27.

In this study, we applied the revised algorithm to produce the NHSnow SD and SWE products. Its original version of the algorithm was published in our previous work (Xiao et al., 2018). Compared to current available published studies in the literature, this study has following improvement and new findings:

1) Compared with the other simple retrieval algorithms stand-only using passive microwave data, our proposed algorithm has great progressive in estimating snow depth due to indirectly considering the evolving of snow grain size and directly distinguish the effect of the different land cover types (Xiao et al., 2018). If the simple SD/SWE retrieval algorithm is applied to the whole snow cover season instead of the proposed algorithm in this study, it would increase the systematic error stemming from the variation and evolution of snow grain size. From the comparison results of different retrieval algorithms, NHSnow product is definitely better than the previous snow depth products (such as: 1. Long-term time series of daily snow depth dataset in China (1979-2018), 2. The snow depth products using NASA algorithm), of which was described in our published work (Xiao et al., 2018).

2) Although GlobSnow products have a better performance in some regions, however, GlobSnow does not cover mountain area, the GlobSnow products are incomplete for investigation of regional and hemispheric-scale snow cover characteristics. By contrast, NHSnow products can provide a daily and full coverage SD and SWE information covering the northern hemisphere. Compared to GlobSnow, NHSnow SD products do a fairly good job in SD estimation and in estimating snow cover information (SD and SWE) under deep snow (SWE$\geq$ 150 mm). Correspondingly, we revised the statement in Section 5 (page 27 lines 1-6)

*"Additionally, we used snow course observation dataset to evaluate the performance of NHSnow and GlobSnow SWE products. The evaluation results indicate that omission error of snow cover identification is one critical source of error for GlobSnow SWE product in the Northern Hemisphere; snow density model could be a major source of error for NHSnow SWE product in Eurasia region. The comparison results between NHSnow and GlobSnow products suggested that NHSnow*

*products have better estimation advantage in deep snow condition (Xiao et al., 2018)."*

3) By analyzing NHSnow SD products, we obtained one interesting finding that statistically significant increases in SD while decreases in SCD during the period 1992-2016 occurring in some regions of the Northern Eurasia, which is coincide with meteorological stations observations.

4) Our analysis results show the winter snow depth experienced the largest variation (-0.11 ± 0.40 cm yr.[-1]) compared to the other two seasons (fall and spring) during 1992-2016; and also the absolute values of the SD variation rate in most polar regions are apparently greater than in the middle-low latitude. We added the figure to the Appendix for the zonal distributions of the seasonal average SD variation rate in three seasons from 1992 to 2016.

[Figure]

Figure D. The zonal distributions of the seasonal average SD variation rate in fall (a), winter (b) and spring (c) from 1992 to 2016. The error bars in (a-c) is one standard deviation from their long-term mean.

Moreover, the literature review presented for SD and SWE retrievals is incomplete. The authors ignore recent results from assimilation of RS data in Land Surface Model, including improved snow model, driven by meteorological data (and/or reanalysis). Such approaches are more interesting given their independent from in-situ snow measurements and provide both SD and SWE data (See Larue et al., 2018, Hydrol. Earth Syst. Sci., 22; Kwon et al. 2016, J. Hy- drometeorol., 17, 2853–2874; Charrois et al., 2016, The Cryosphere, 10:1021–1038; De Lannoy et al., 2012, Water Resour. Res., 48, W01522). Also recent active PM SAR-based analysis can provide SD data at high spatial resolution : coherence analysis (Singh et al., Water 2020,

12, 21) or phase difference from ESA Sentinel constellation, Leinss, S.; Parrella, G.; Hajnsek, I. Snow height determination by polarimetric phase differences in X-band SAR data. IEEE J. Sel. Top. Appl. Earth Observ. Remote Sens. 2014, 7, 3794–3810), also completely independently from in-situ data!

Response: Thanks for your suggestion. In introduction section, we added the literature review about SD and SWE retrieval by citing related literature in page 2 lines 26-31.

*"In addition to directly using passive microwave brightness temperature to retrieve SD/SWE, there are several methods that have been developed to obtain SWE and SD estimates by assimilating the brightness temperature into a snow physical model or/and radiative transfer model (Takala et al., 2011; Kwon et al., 2016; Larue et al., 2018). Gradually, the other sources of data were similarly applied to assimilation model as auxiliary information to improve SD/SWE estimates, for example, optical reflectance (Charrois et al., 2016), snow cover fraction(De Lannoy et al., 2012; Toure et al., 2018)"*

In their paper, the authors analyzed also the trend of SD (mean and max), SWE, Snow Cover Extent (SCE) and Snow Cover Duration (SCD), showing similar known results than those already published. There are no really new insights here, even if the results are well presented with maps showing spatial variability between North Hemisphere regions (excepted trends slighted over too short periods, see bellow). Also, the authors do not discuss the fact that results based on dry snow only are biased in spring when snow is generally wet. Finally, this paper brings any explanation on the observed trends (some period and areas with increase or decrease snow parameters), as the authors recognized at the end of the paper.

Overall, I recognize that to produce a global dataset is a strong work and that the authors succeed to reach the mean accuracy level of existing databases, but this paper is relatively weak in its original scientific contribution (any real improvement; trends more or less known). I thus don't recommend its publication in TC.

This paper describing the NHsnow database should be submitted to the dedicated journal for new released datasets: Earth Syst. Sci. Data.

Response: Thanks for your suggestions. As the literature review and description in Section 1 (page 4 lines 1-6),

*"However, most of published studies concerned the regional area (Russia or Canada) giving a detailed description and analysis of snow cover variation characteristics (Bulygina et al., 2011;Brown et al., 2019), or concentrated on the northern hemisphere with the limited description of snow cover change characteristics (i.e. only one or two snow cover variables was/were used to analyze), resulting in that they did not address changes characteristics of snow depth (Wu et al., 2018;Mudryk et al., 2015). Additionally, there is no one assume that getting information on the trend of one snow cover characteristic implies knowing the variation trends of other characteristics".*

Therefore, one aim of current work is

*"... to provide a comprehensive changes characteristics description of snow cover in three major snow characteristics (snow depth, snow water equivalent, and snow cover duration) since 1992 to 2016"* described in page 4 lines 18-20.

Apart from providing comprehensive information related to snow cover variation in northern hemisphere, current study also have one finding described in "Conclusion" section in page 27 lines 20-23

"*It is worthwhile to note that statistically significant increases in average SD in winter while decreases in SCD during the period 1992-2016 occurring in some regions of the North Russia that is coincide with the results of Bulygina et al. (2011) and Zhong et al. (2018) using meteorological stations observations.*"

As for the discussion on wet snow, we added the description information in page 21 lines 23-27

"*We have to note that snow-covered area of NHSnow product was dry snow determined by Grody's series of rules (Grody and Basist, 1996). The snowpack inevitably contains liquid water since snow melting events occur especially in spring and summer. During spring and summer periods, consequently, the snow mass and SCD derived from NHSnow product will be partly underestimated and the bias may occur in the change trend analysis of snow cover*"

Specific comments

1. Introduction: incomplete literature review about other approaches. Also, limitations of SWE retrieval based on PM are not well reviewed. One of the main problem is the snow microstructure (grain size, stratigraphy, ice crust layer: : :) that evolves during the winter and that strongly affects the PM emission, more than SWE! (see Sandells et al.,2017, The Cryosphere, 11, 229–246; Roy et al., 2016, The Cryosphere, 10; Durand et al., 2011, IEEE Geosci. Remote Se., 8 ; : : : and Matzler, 1987, Remote Sens. Rev., 2, 259–387).

Response: Thank you for your suggestion. We added the literature review about the effect of snow microstructure in SD and SWE retrieval in page 3 lines 9-12

"*The other explanation is that the effect of snow microstructure evolution (grain size, stratigraphy) (Durand et al., 2011). The evolution of snow microstructure have strong effect on the upwelling microwave radiation through and emitted from snowpack (Sandells et al., 2017;Roy et al., 2016); in other word, the improvement of SD/SWE retrieval would be benefit from providing the more detailed snow microstructural parameters (Dai et al., 2012).*"

3.3 Estimation of SWE Very empirical approach (Eq. 3 and Table 3), and without statistical error analysis? PM data are known to be limited over deep snow (see Larue et al., 2017, Remote Sens. Environ., 194).

Response: Thanks for your comment. The snow density model indeed is a statistical method with good applicability in Northern America (NA) area, and its validation and evaluation also mainly focus on NA, especially in Canada (Hill et al, 2019; Strum, et al, 2011). SWE is determined by snow density and SD. That is to say that the snow density method used may bring error in SWE estimation. We added the validation and evaluation of SWE using in-situ observation dataset during 1992-2014 in the updated manuscript. We evaluated and analyzed the effect of deep snow (SWE ≥ 150 mm) and shallow snow (SWE < 150 mm) in SWE product. (Detailed analysis description is gave Section 4.1.2). In this study, the SWE retrieval is

based on an empirical density equation. Based on our analysis, we found that this statistical snow density model used in NHSnow SWE product may be a major error source in calculating SWE from SD, especially in Eurasia area.

Reference:

Hill, D. F., et al. : Converting snow depth to snow water equivalent using climatological variables, The Cryosphere, 13, 1767-1784, 10.5194/tc-13-1767-2019, 2019.

Sturm, M., et al. : Estimating snow water equivalent using snow depth data and climate classes, Journal of Hydrometeorology, 11, 1380-1394, 2010.

In term of deep snow, we added the discussion about the deep snow and saturation effect at specific frequency in Section 4 (Results and Discussion)

*"In addition to above discussed factors, deep snow is another critical influence factor in SD and SWE retrieval. Because of the saturation of the penetration depth at 37 GHz, many studies indicate that deep snow is a major source of uncertainties in SD and SWE retrieval when using passive microwave brightness temperature (Roy et al., 2016;Durand et al., 2011;Larue et al., 2017;Saberi et al., 2019)."* In page 19, lines 10-13.

4. Results Yes, in-situ SWE datasets exist for data over Siberia (Bulygina, O., Groisman, P. Y., Razuvaev, V., and Korshunova, N. (2011). Changes in snow cover characteristics over northern eurasia since 1966. Environmental Research Letters, 6(4):045204) and over Canada (Brown, R. D., Fang, B., and Mudryk, L. (2019). Update of canadian historical snow survey data and analysis of snow water equivalent trends, 1967-2016: Research note. Atmosphere-Ocean, 1-8).

Response: Thank you very much. As provided by Reviewer 1#, there is a project (ERA-CLIM2) that have compiled the SWE observation records over Norther Hemisphere covering Russia, Canada and Finland (Section 2.2 show the detailed information about this data project). Some of these datasets over Russia and Canada were used in above mentioned two publication. In Section 2.2, we added the dataset description information in page 6 lines 4-10,

*"To evaluate the performance of SWE product, the in-situ SWE datasets were collected and utilized in this work. The ERA-CLIM2 data set is available (http://litdb.fmi.fi/eraclim2.php) providing northern hemisphere snow course observations (including SWE, SD, snow bulk density). This dataset contains more than 958 000 records spanning from 1935 to 2014 and mainly distributed in the former Soviet Union, Canada, and Finland. From this dataset, we selected 1331 measurement sites of which the records is since 1992 (Fig. 2) for SWE validation.*

[Figure]

*Figure 2. Spatial distribution of the in-situ sites with SWE in Northern Hemisphere.*

"

This snow course dataset was used to validate the SWE product (NHSnow and GlobSnow) in Section 4.1.2. In this section, we exhibited the validation and evaluation of SWE and gave the detailed error source analysis.

*"To perform the SWE evaluation, we acquired more than 77 000 valid data records of NHSnow, GlobSnow, and in-situ measurements from 1992 to 2014 (December to February). We conducted performance for both NHSnow and GlobSnow SWE products in shallow (< 150 mm) and deep (≥ 150 mm) snow conditions. (Larue et al., 2017). The performance metrics were summarized in Table 5 for both NHSnow and GlobSnow SWE products against snow course observation over the former Soviet Union, Canada, and Finland. The overall bias, MAE and RMSE for NHSnow SWE products are 43.6 mm, 61.9 mm and 87.3 mm respectively; while for GlobSnow, they are 15.3 mm, 31.6 mm, 61.5 mm respectively. For shallow snow condition (SWE < 150 mm), the bias, MAE and RMSE are slightly reduced for both SWE products compared to using total records condition (Table 5). Nevertheless, when analyzing the deep SWE (≥ 150 mm), the statistics results for both SWE products are relative large (NHSnow: bias = -46.0 mm, MAE = 103.6 mm, RMSE = 169.0 mm; GlobSnow: bias = -103.1 mm, MAE = 112.2 mm, RMSE = 201.0 mm; Table 5). In general, the GlobSnow SWE products have a better performance than NHSnow SWE product, especially in shallow snow condition (< 150 mm). However, for deep snow, NHSnow SWE product has a better performance, with less bias, MAE, and RMSE, than GlobSnow SWE product.*

*The evaluation performance of both NHSnow and GlobSnow SWE products was conducted with respect to SWE in-situ observations over the former Soviet Union, Canada, and Finland (Fig. 9). For GlobSnow SWE product, the SWE estimation have the best performance in the former Soviet Union region area than the other two regions (Canada and Finland) with the least bias and MAE; For GlobSnow SWE products, the estimation performance in Finland region which with less bias, while MAE and RMSE is better than that in Canadian regions (Fig. 9). Our analysis result is consistent with the accuracy evaluation results from previous published study in Canadian regions (Larue et al., 2017). Through analyzing the SWE in-situ observations, we found that omission error of snow cover identification, which means that in-situ observation is fully snow-covered while the prediction of snow cover algorithm is snow-free, is the main error source for GlobSnow SWE product. The omission error of GlobSnow SWE product in the former Soviet Union, Canada, and Finland are 6.1%, 18.4 and 8.2% respectively. As for NHSnow SWE product, the best estimation performance is in Canada; then in the former Soviet Union;*

*the estimated SWE in Finland was not as good as the regions (Fig. 9). Unlike GlobSnow, the NHSnow SWE products do not have the omission error but the commission error, which is defined as snow-free observed by in-situ data but snow-covered detected by snow cover algorithm. The commission errors for NHSnow product are 0, 8.9%, and 0 for the former Soviet Union, Canada, and Finland respectively. Though the commission error is 8.9% in Canada, the NHSnow SWE estimation performance (less bias and MAE; -7.9 and 48.9) is better than in the other two regions. It may be due to the statistical model of snow density, which is obtained through the snow observation data across North America (Sturm et al., 2010). In other words, this snow density model has a better applicability in North America (Hill et al., 2019), but may not be the case in the other two regions. The large RMSE in Canada than in the former Soviet Union may be due to deep snow because there is 10.4% of deep snow in Canada. Snow density may also contribute to misestimating SWE. For NHSnow SWE product in the former Soviet Union region with 6.1% deep snow records, there is no commission error and the error of the estimated snow depth is relatively small (bias < 5 cm, MAE < 15 cm from Section 4.1.1), we thus assume that the errors in SWE primarily come from the modeled snow density in the former Soviet Union region. Through analyzing SWE observation and SD estimates of NHSnow, the performance in Finland may be accounted for by two most possible reasons which are deep snow (more than 8% records is deep snow) and the inaccurate estimates of SD (bias = -13.3, MAE = 18.3 from Section 4.1.1). Additionally, there are no commission error in Finland region; therefore, the inaccurate SD estimates would be the main error source of NHSnow SWE product in Finland region.*

*Based on the above analysis, NHSnow does a fairly good work in SD estimation (bias = -0.59 cm, MAE = 15.98 cm, and RMSE = 20.11 cm). Although deep snow is a great challenge for current SWE products, NHSnow SWE product have less error in deep snow compared to GlobSnow SWE product (Table 5, Fig. 9). Moreover, the statistical snow density model proposed by Sturm et al. (2010) cannot accurately describe the evolution of snow density in Eurasia region, and it may be a major source of error for NHSnow SWE product, which needs further investigation when related data are available.*

*Table 5. Summary of performance indexes (Bias, MAE, RMSE; unit: mm) for NHSnow SWE product and GlobSnow SWE product against SWE in-situ measurements during 1992 to 2014 (December–February)*

|  | Total | | < 150 mm | | ≥ 150 mm | |
|---|---|---|---|---|---|---|
|  | GlobSnow | NHSnow | GlobSnow | NHSnow | GlobSnow | NHSnow |
| Bias | 15.3 | 43.6 | 11.0 | 41.4 | -103.1 | -46.0 |
| MAE | 31.6 | 61.9 | 25.6 | 59.9 | 112.2 | 103.6 |
| RMSE | 61.5 | 87.3 | 32.8 | 77.8 | 201.0 | 169.7 |

[Figure]

*Figure 9. The performance evaluation of two SWE products (NHSnow and GlobSnow) with respect to SWE in-situ observations over the former Soviet Union, Canada, and Finland using three indexes (Bias, MAE, RMSE; unit: mm)"*

All the maps are too small, hard to read. Seasonal trend analysis biased when based on dry snow. Have you eliminated wet snow from ERAi outputs?

Response: Thanks for your comment. The figures (Figure 8, 11; Figure A-C in the Appendix) in this paper were updated. In snow cover variation characteristic analysis section, we did not utilize the ERA-Interim/Land data. The wet snow exists in the whole snow season when melting event occurring, especially in spring (from March to June). The proposed snow cover product (NHSnow) used some detection rules to eliminate the wet snow. As you say, seasonal trend analysis biased when based on dry snow. We added the discussion description in page 21 lines 23-27

"*We have to note that snow-covered area of NHSnow product was dry snow determined by Grody's series of rules (Grody and Basist, 1996). The snowpack inevitably contains liquid water since snow melting events occur especially in spring and summer. During spring and summer periods, consequently, the snow mass and SCD derived from NHSnow product will be partly underestimated and the bias may occur in the change trend analysis of snow cover*"

4.2 Snow mass trend I don't agree with the snow mass trend over too short periods (1992-2001) and 2002-2016) (Fig. 10). A trend over only 10 years makes no sense: you only change one value in the series, and the slope changes drastically! Such analysis has no interest here (maybe for sensationalism public journals!) Analysis of SWE is insufficient.

Response: Thanks for your comments and suggestions. For period (1992-2001), we do know that it is really short for variation trend analysis. One goal of this work is to explore the snow cover variation characteristic since the new century; thus, we mainly concentrate on analyzing snow cover characteristics during 2002-2016 period. In previous version of the manuscript only have two periods (1992-2016; 2000-2016), one reviewer once gave me a comment on this point that the analysis periods

should be complete even if it is short or not pass significance test. Therefore, we modified the analysis period and perform the variation characteristic of snow cover in following three analysis periods, including the short period (1992-2001), the longer period (2002-2016) and the whole period (1992-2016), in Section 4.2 (Variation of snow depth) and Section 4.3 (Snow mass). Because the ten years (1992-2001) is very short and most of the results in this period did not pass significance test, we did not give more analysis on these results that only used to be as reference.

As to SWE evaluation and analysis, it was added to Section 4.1.2, in which we gave more detailed comparison analysis about two SWE products (NHSnow and GlobSnow). We do know that detailed analysis of SWE is necessary and the analysis of SWE in current study is somewhat inadequate. As described in "Space distribution of long-term means of maximum snow water equivalent follows in many respects the distribution of maximum snow depth" (Bulygina et al, 2011), therefore, we did not conduct the spatiotemporal analysis of maximum SWE like the analysis of SD. Moreover, as written at the end of Section 6 (Conclusion), future work will involve more detailed analysis of SWE:

*"So far, further analyses and study are still need to help us to deeply understand the changes of SWE in the northern hemisphere, e.g. analyzing the difference of snow mass variation and its response to climate change in two major continents (Eurasia and North America) (Takala et al., 2011; Jeong et al., 2016) and investigating the variation trends of the peak of SWE in response to climate change in regional or hemispheric regions (Irannezhad et al., 2016; Musselman et al., 2017; Brown and Mote, 2009; Zeng et al., 2018)."*

Reference:

Bulygina, O., Groisman, P. Y., Razuvaev, V., and Korshunova, N. (2011). Changes in snow cover characteristics over northern eurasia since 1966. Environmental Research Letters, 6(4):045204

4.3 Snow cover days: the usually term used is "Snow Cover Duration" (SCD)

Response: Thanks. The term "snow cover days" changed to "*snow cover duration*".

5. Conclusion No convincing arguments for using NHsnow instead of others? (added value?, improvements?).

Response: Thanks for your comment. In this study, we applied the revised algorithm to produce the NHSnow SD and SWE products. Its original version of the algorithm was published in our previous work (Xiao et al., 2018). Compared to current available published studies in the literature, this study has following improvement and new findings:

1) Compared with the other simple retrieval algorithms stand-only using passive microwave data, our proposed algorithm has great progressive in estimating snow depth due to indirectly considering the evolving of snow grain size and directly distinguish the effect of the different land cover types (Xiao et al., 2018). If the simple SD/SWE retrieval algorithm is applied to the whole snow cover season instead of the proposed algorithm in this study, it would increase the systematic error stemming from the variation and evolution of snow grain size. From the comparison results of different retrieval algorithms,

NHSnow product is definitely better than the previous snow depth products (such as: 1. Long-term time series of daily snow depth dataset in China (1979-2018), 2. The snow depth products using NASA algorithm), of which was described in our published work (Xiao et al., 2018).

2) Although GlobSnow products have a better performance in some regions, however, GlobSnow does not cover mountain area, the GlobSnow products are incomplete for investigation of regional and hemispheric-scale snow cover characteristics. By contrast, NHSnow products can provide a daily and full coverage SD and SWE information covering the northern hemisphere. Compared to GlobSnow, NHSnow SD products do a fairly good job in SD estimation and in estimating snow cover information (SD and SWE) under deep snow (SWE $\geq$ 150 mm). Correspondingly, we revised the statement in Section 5 (page 27 lines 1-6):

*"Additionally, we used snow course observation dataset to evaluate the performance of NHSnow and GlobSnow SWE products. The evaluation results indicate that omission error of snow cover identification is one critical source of error for GlobSnow SWE product in the Northern Hemisphere; snow density model could be a major source of error for NHSnow SWE product in Eurasia region. The comparison results between NHSnow and GlobSnow products suggested that NHSnow products have better estimation advantage in deep snow condition (Xiao et al., 2018)."*

3) By analyzing NHSnow SD products, we obtained one interesting finding that statistically significant increases in SD while decreases in SCD during the period 1992-2016 occurring in some regions of the Northern Eurasia, which is coincide with meteorological stations observations.

4) Our analysis results show the winter snow depth experienced the largest variation (-0.11 $\pm$ 0.40 cm yr.$^{-1}$) compared to the other two seasons (fall and spring) during 1992-2016; and also the absolute values of the SD variation rate in most polar regions are apparently greater than in the middle-low latitude. We added the figure to the Appendix for the zonal distributions of the seasonal average SD variation rate in three seasons from 1992 to 2016.

[Figure]

Figure D. The zonal distributions of the seasonal average SD variation rate in fall (a), winter (b) and spring (c) from 1992 to 2016. The error bars in (a-c) is one time of standard deviation.

Of course, data assimilation is the most promising method and has an incomparable with other method simplified the physics and without error correction and uncertainty involving. Learn about the disadvantages and advantages of our snow cover products (NHSnow) compared to the best snow products, so as to facilitate us to continue to improve the snow depth and snow water equivalent retrieval algorithm and the snow cover product in subsequent work. Based on the comparison results, it inspires us that combining the data assimilation method and the machine learning method would be a promising field to improve snow depth and snow water equivalent retrieval accuracy. We revised the description in the Conclusion section (page 27 lines 26-30):

"*Comparing to the current relative best snow water products, there are some deficiencies and limitations (e.g. overestimation, underestimation) for NHSnow products, further efforts should be made to improve the estimation accuracy and robustness of the SD inversion algorithm (especially for SWE products) coupling snow physical model and/or radiative transfer model that appears to be a promising approach (Xue et al., 2018; Larue et al., 2018; Larue et al., 2017).*"